# Boosting the Confidence of Near-Tight Generalization Bounds for Uniformly Stable Randomized Algorithms

## Abstract

High probability generalization bounds of uniformly stable learning algorithms have recently been actively studied with a series of near-tight results established by Feldman & Vondrak (2019); Bousquet et al. (2020). However, for randomized algorithms with on-average uniform stability, such as stochastic gradient descent (SGD) with time decaying learning rates, it still remains less well understood if these deviation bounds still hold with high confidence over the internal randomness of algorithm. This paper addresses this open question and makes progress towards answering it inside a classic framework of confidence-boosting. To this end, we first establish an in-expectation first moment generalization error bound for randomized learning algorithm with on-average uniform stability, based on which we then show that a properly designed subbagging process leads to near-tight high probability generalization bounds over the randomness of data and algorithm. We further substantialize these generic results to SGD to derive improved high probability generalization bounds for convex or non-convex optimization with natural time decaying learning rates, which have not been possible to prove with the existing uniform stability results. Specially for deterministic uniformly stable algorithms, our confidence-boosting results improve upon the best known generalization bounds in terms of a logarithmic factor on sample size, which moves a step forward towards resolving an open question raised by Bousquet et al. (2020).

## 1  Introduction

In many statistical machine learning problems, the ultimate goal is to design a suitable algorithm $A : \mathcal{Z}^N \mapsto \mathcal{W}$ that maps a training data set $S = \{z_i\}_{i \in [N]} \in \mathcal{Z}^N$ to a model $A(S)$ in a closed subset $\mathcal{W}$ of an Euclidean space such that the following population risk function evaluated at the model is as small as possible:
$$R(A(S)) := \mathbb{E}_{Z \sim \mathcal{D}}[\ell(A(S); Z)],$$
where $\ell : \mathcal{W} \times \mathcal{Z} \mapsto \mathbb{R}^+$ is a non-negative bounded loss function whose value $\ell(w; z)$ measures the loss evaluated at $z$ with parameter $w$, and $\mathcal{D}$ represents a distribution over $\mathcal{Z}$. It is generally the case that the underlying data distribution is unknown, and in this case the training data is usually assumed to be an i.i.d. set, i.e., $S \overset{\text{i.i.d.}}{\sim} \mathcal{D}^N$. Then, a natural alternative measurement that mimics the computationally intractable population risk is the empirical risk defined by

$$R_S(A(S)) := \mathbb{E}_{Z \sim \text{Unif}(S)}[\ell(A(S); Z)] = \frac{1}{N} \sum_{i=1}^{N} \ell(A(S); z_i).$$

The bound on the difference between population and empirical risks is of central interest in understanding the generalization performance of learning algorithm $A$. Particularly, we hope to derive a suitable law of large numbers, i.e., a sample size vanishing rate $b_N$ such that the generalization bound $|R_S(A(S)) - R(A(S))| \lesssim b_N$ holds with high probability over the randomness of $S$ and potentially the randomness of $A$ as well. Provided that $A(S)$ is an almost minimizer of the empirical risk function $R_S$, say $R_S(A(S)) \lesssim \min_{w \in \mathcal{W}} R_S(w) + b_N$, the generalization bound immediately implies an *excess risk* bound $R(A(S)) - \min_{w \in \mathcal{W}} R(w) \lesssim b_N + \frac{1}{\sqrt{N}}$ due to the standard risk decomposition and Hoeffding's inequality. Therefore, generalization bounds also play a crucial role in understanding the stochastic optimization performance of a learning algorithm.

The generalization bounds can be naturally implied by uniform bounds on $\sup_{w \in \mathcal{W}} |R(w) - R_S(w)|$ (Bartlett et al., 2006; Bottou & Bousquet, 2008). While broadly applicable (e.g., to non-convex problems) and leading to tight generalization in some specific regimes (e.g., margin-based learning (Kakade et al., 2009)), uniform convergence bounds in general cases might suffer from the polynomial dependence on dimensionality and thus are not suitable for high-dimensional models which are ubiquitous in modern machine learning. Alternatively, a powerful proxy for analyzing the generalization bounds is the *stability* of learning algorithms to changes in the training dataset. Since the seminal work of Bousquet & Elisseeff (2002), stability has been extensively demonstrated to beget dimension-independent generalization bounds for deterministic learning algorithms (Mukherjee et al., 2006; Shalev-Shwartz et al., 2010), as well as for randomized learning algorithms (such as bagging and SGD) (Elisseeff et al., 2005; Hardt et al., 2016). So far, the best known results about generalization bounds are offered by approaches based on the notion of uniform stability (Feldman & Vondrak, 2018; 2019; Bousquet et al., 2020). Inspired by these recent breakthrough results, we seek to derive sharper high-probability generalization bounds for randomized learning algorithms with on-average uniform stability. A concrete working example of our study is the widely used stochastic gradient descent (SGD) algorithm that carries out the following recursion for all $t \geq 1$ with learning rate $\eta_t > 0$:

$$w_t := \Pi_{\mathcal{W}} \left( w_{t-1} - \eta_t \nabla_w \ell(w_{t-1}; z_{\xi_t}) \right), \tag{1}$$

where $\xi_t \in [N]$ is a uniform random index of data under with or without replacement sampling, and $\Pi_{\mathcal{W}}$ is the Euclidean projection operator associated with $\mathcal{W}$. Despite its popularity in the study of stability theory (Hardt et al., 2016; Zhou et al., 2019; Lei & Ying, 2020), the high probability generalization bounds of SGD are still relatively under explored through the lens of uniform stability.

## 1.1 PRIOR RESULTS

Let us now briefly review some state-of-the-art generalization bounds for uniformly stable algorithms and their randomized variants. We denote by $S \doteq S'$ if a pair of data sets $S$ and $S'$ differ in a single data point. A randomized learning algorithm $A$ is said to have on-average $\gamma_N$-uniform stability if it satisfies the following uniform bound

$$\sup_{S \doteq S', z \in \mathcal{Z}} |\mathbb{E}_A[\ell(A(S); z)] - \mathbb{E}_A[\ell(A(S'); z)]| \leq \gamma_N.$$

This definition is equivalent to the concept of uniform stability defined over the on-average loss $\mathbb{E}_A[\ell(A(S); z)]$. Suppose that the loss function is Lipschitz and bounded in the interval $(0, 1]$. Then essentially it has been shown in Feldman & Vondrak (2019) that for any $\delta \in (0, 1)$, with probability at least $1 - \delta$ over $S$, the on-average generalization error is upper bounded by

$$|\mathbb{E}_A[R(A(S)) - R_S(A(S))]| \lesssim \gamma_N \log(N) \log\left(\frac{N}{\delta}\right) + \sqrt{\frac{\log(1/\delta)}{N}}. \tag{2}$$

Recently, Bousquet et al. (2020) derived a slightly improved uniform stability bound that implies

$$|\mathbb{E}_A[R(A(S)) - R_S(A(S))]| \lesssim \gamma_N \log(N) \log\left(\frac{1}{\delta}\right) + \sqrt{\frac{\log(1/\delta)}{N}}. \tag{3}$$

These generalization bounds are near-tight (up to a logarithmic factor $\log(N)$) in the sense of a lower bound on sum of random functions provided in that paper. While sharp in the dependence on sample size, one common limitation of the above uniform stability implied generalization bounds lies in that these high-probability results only hold *in expectation* with respect to the internal randomness of algorithm.

Further suppose that $A$ has $\gamma_N$-uniform stability with probability at least $1 - \delta'$ for some $\delta' \in (0, 1)$ over the randomness of $A$, i.e.,

$$\mathbb{P}_A \left\{ \sup_{S \doteq S', z \in \mathcal{Z}} |\ell(A(S); z) - \ell(A(S'); z)| \leq \gamma_N \right\} \geq 1 - \delta'. \tag{4}$$

Suppose that the randomness of $A$ is independent of the training set $S$. Then with probability at least $1 - \delta - \delta'$ over $S$ and $A$, the bound of Bousquet et al. (2020) naturally implies

$$|R(A(S)) - R_S(A(S))| \lesssim \gamma_N \log(N) \log\left(\frac{1}{\delta}\right) + \sqrt{\frac{\log(1/\delta)}{N}}. \tag{5}$$

---

**Algorithm 1:** `Confidence-Boosting for Randomized Learning`

---

**Input** : A randomized learning algorithm $A$ and a training data set $S = \{z_i\}_{i \in [N]} \overset{\text{i.i.d.}}{\sim} \mathcal{D}^N$.

**Output**: $A_{k^*}(S_{k^*})$.

Uniformly divide $S$ into $K$ *disjoint* subsets such that $S = \bigcup_{k \in [K]} S_k$ and $|S_k| = \frac{N}{K}, \forall k \in [K]$.

**for** $k = 1, 2, ..., K$ **do**

| Estimate $A_k(S_k)$ as an output of the randomized algorithm $A$ over subset $S_k$.

**end**

Compute $k^* = \arg\min_{k \in [K]} \left| R_{S \setminus S_k}(A_k(S_k)) - R_{S_k}(A_k(S_k)) \right|$.

---

This is by far the best known generalization bound of randomized stable algorithms that hold with high probability jointly over data and algorithm. The result, however, relies heavily on the high-probability uniform stability condition expressed in equation 4. For the SGD (see equation 1) with fixed learning rate $\eta_t \equiv \eta$, it is possible to show that $\gamma_N \lesssim \eta\sqrt{T} + \frac{\eta T}{N}$ and $\delta' = N \exp(-\frac{N}{2})$ in equation 4 (Bassily et al., 2020). For SGD with time decaying learning rate that has been widely studied in theory Harvey et al. (2019); Rakhlin et al. (2012) and applied in practice for training popular deep nets such as ResNet and DenseNet Bengio et al. (2017), it is not clear if the condition in equation 4 is still valid for $\gamma_N$ and $\delta'$ of interest. On the other hand, it is possible to show (see the proofs of Corollary 1 and 2) that SGD with time decaying learning rate has desirable on-average uniform stability.

More specially for randomized learning methods such as bagging (Breiman, 1996) and SGD equation 1, the randomness of algorithm can be precisely characterized by a vector of i.i.d. parameters $\xi = \{\xi_1, ..., \xi_T\}$ which are independent on data $S$. In such cases, suppose that $A(S; \xi)$ has uniform stability with respect to $\xi$ at any given $S$, i.e., $\sup_{\xi \doteq \xi'} |\ell(A(S; \xi)) - \ell(A(S; \xi'))| \leq \rho_T$. Then the high probability bound established in Elisseeff et al. (2005) shows that with probability at least $1 - \delta$,

$$|R(A(S)) - R_S(A(S))| \lesssim \gamma_N + \left( \frac{1 + N\gamma_N}{\sqrt{N}} + \sqrt{T}\rho_T \right) \sqrt{\log\left(\frac{1}{\delta}\right)}. \qquad (6)$$

Provided that $\gamma_N \lesssim \frac{1}{N}$ and $\rho_T \lesssim \frac{1}{T}$, the above bound shows that the generalization bound scales as $\mathcal{O}(\frac{1}{\sqrt{N}} + \frac{1}{\sqrt{T}})$ with high probability. However, the above bound will show no guarantee on convergence if $\gamma_N \gtrsim \frac{1}{\sqrt{N}}$ and/or $\rho_T \gtrsim \frac{1}{\sqrt{T}}$. For example, this is actually the case for SGD with time decaying learning rate $\eta_t = O(\frac{1}{t})$ on non-convex loss functions in which $\gamma_N \lesssim \frac{\sqrt{T}}{N}$ and $\rho_T$ can scale as large as $\mathcal{O}(1)$.

**Open problem and motication.** Keeping the merits and deficiencies of above recalled prior results in mind, it still remains an open issue if the existing deviation bounds can hold with high confidence for randomized algorithms with on-average uniform stability (such as SGD with decaying learning rates). The goal of the present study is to derive sharper high-probability generalization bounds for randomized algorithms that hold jointly over the randomness of data and algorithm, based on the relatively weaker notion of on-average uniform stability rather than its high probability counterpart.

## 1.2 OVERVIEW OF OUR RESULTS

The confidence-boosting technique of Schapire (1990) is a classical meta approach that allows us to boost the dependence of a learning algorithm on the failure probability $\delta$ from $1/\delta$ to $\log(1/\delta)$, at a certain cost of computational complexity. The fundamental contribution of our work is to reveal that the confidence-boosting trick yields near-tight high probability generalization bounds for uniformly stable randomized learning algorithms. The novelty lies in a refined analysis of the in-expectation first moment generalization error bound for a randomized learning algorithm with on-average uniform stability, which leads to improved high-probability generalization bounds over the randomness of data and algorithm via confidence-boosting. More specifically, given a randomized learning algorithm $A$, we consider the following subbagging process over training set:

**Boosting the Confidence via Subbagging**. We independently run $A$ over $K$ *disjoint* and u-niformly divided training subsets $\{S_k\}_{k \in [K]}$ to obtain solutions $\{A_k(S_k)\}_{k \in [K]}$. Then we e-

valuate the validation error of each candidate solution over its complementary training subset, and output $A_{k^*}(S_{k^*})$ that has the smallest gap between training error and validation error, i.e., $k^* = \arg\min_{k \in [K]} \left| R_{S \backslash S_k}(A_k(S_k)) - R_{S_k}(A_k(S_k)) \right|$. Specially when $A$ is deterministic, this reduces to a standard subbagging process, namely a variation of bagging using without-replacement sampling for subsets generation (see, e.g., Andonova et al., 2002). In general, this is essentially a subbagging procedure with greedy model selection for randomized algorithms over multiple disjoint subsets. Here we have assumed without loss of generality that $N$ is a multiplier of $K$. The considered procedure of confidence-boosting for randomized learning is outlined in Algorithm 1.

**Main Results.** In what follows, we highlight our main results on the generalization bounds of the output of Algorithm 1 along with the implications for SGD and deterministic algorithms:

- *General results.* Suppose that the loss is Lipschitz and bounded in the range of $(0, 1]$. Our main result in Theorem 1 show that for any $\delta \in (0, 1)$, setting $K \asymp \log(\frac{1}{\delta})$ yields the following generalization bound of the output of Algorithm 1 that holds with probability at least $1 - \delta$ over $S$ and $\{A_k\}_{k \in [K]}$:

$$|R(A_{k^*}(S_{k^*})) - R_S(A_{k^*}(S_{k^*}))| \lesssim \frac{1}{K}\left(\sqrt{\gamma_{\mathrm{m}^2, \frac{N}{K}}} + \gamma_{\mathrm{m}, \frac{N}{K}} + \sqrt{\frac{K}{N}}\right) + \sqrt{\frac{\log(K/\delta)}{N}},$$

  where $\gamma_{\mathrm{m}, N}$ and $\gamma_{\mathrm{m}^2, N}$ are respectively mean-uniform stability and mean-square-uniform stability bounds introduced in Definition 1. In contrast to the bound in equation 6, our bound is not relying on the uniform stability with respect to the random bits of algorithm.

- *Stronger generalization bounds for SGD via confidence-boosting.* We then use our general results to study the benefit of confidence-boosting on the generalization bounds of SGD-w (SGD via with-replacement sampling as outlined in Algorithm 2). The main results are a series of corollaries of Theorem 1 when substantialized to SGD with smooth (Corollary 1) or non-smooth (Corollary 2) convex loss, and smooth non-convex loss functions (Corollary 3). For an instance, our result in Corollary 1 showcases that when invoked to SGD-w on smooth convex loss with learning rates $\eta_t = \mathcal{O}(\frac{1}{\sqrt{t}})$, the generalization bound of the output of Algorithm 1 with $K \asymp \log(\frac{1}{\delta})$ is upper bounded by

$$|R(A_{\text{SGD-w}, k^*}(S_{k^*})) - R_S(A_{\text{SGD-w}, k^*}(S_{k^*}))| \lesssim \sqrt{\frac{\log(T)}{N}} + \frac{\sqrt{T} + \sqrt{N}\log(1/\delta)}{N}.$$

  Compared with the $\mathcal{O}(\frac{\sqrt{T}}{N})$ in-expectation bound of smooth convex SGD (Hardt et al., 2016), our above bound is competitive in order while it holds with high confidence.

- *Sharper bounds for uniformly stable algorithms via confidence-boosting.* Specially for a deterministic learning algorithm $A$ that has $\gamma_N$-uniform stability for each data set size $N \geq 1$, we further show through Corollary 4 that the following bound holds for the output of Algorithm 1 with $K = \log(\frac{1}{\delta})$:

$$|R(A(S_{k^*})) - R_S(A(S_{k^*}))| \lesssim \gamma_{\frac{N}{K}} + \sqrt{\frac{\log(K/\delta)}{N}}.$$

  In the case of $\gamma_N \lesssim \frac{1}{\sqrt{N}}$ which holds in some popular learning paradigms such as regularized ERM, the above bound implies (recall $K \asymp \log(\frac{1}{\delta})$)

$$|R(A(S_{k^*})) - R_S(A(S_{k^*}))| \lesssim \sqrt{\frac{\log(1/\delta)}{N}},$$

  which is sharper than the best known result in equation 5 (under $\delta' = 0$) in the sense of the removal of a $\log(N)$ factor. This is a side contribution of our work that might be of independent interest towards answering an open question raised by Bousquet et al. (2020).

In addition to the generalization bounds, we have also derived a high probability excess risk bound for uniformly stable randomized learning with confidence-boosting. More specifically, with a proper modification of the output of Algorithm 1, we can show that with probability at least $1 - \delta$ over $S$ and $\{A_k\}_{k \in [K]}$:

$$R(A_{k^*}(S_{k^*})) - \min_{w \in \mathcal{W}} R(w) \lesssim \gamma_{\mathrm{m}, \frac{N}{K}} + \Delta_{\text{opt}} + \sqrt{\frac{\log(K/\delta)}{N}},$$

where $\Delta_{\text{opt}}$ is the in-expectation empirical risk optimization error as given by equation 7.

## 2 GENERALIZATION ANALYSIS WITH CONFIDENCE-BOOSTING

In this section, we present a set of generic results on the generalization bounds of randomized learning algorithms with confidence-boosting as described in Algorithm 1.

### 2.1 PRELIMINARIES AND A KEY LEMMA

We first introduce the following concept of mean-uniform stability that serves as a powerful tool for analyzing the generalization bounds of randomized learning algorithms (Hardt et al., 2016; Elisseeff et al., 2005; Bassily et al., 2020).

**Definition 1** (Mean and Mean-Square Uniform Stability of Randomized Algorithms). *Let $A$ : $\mathcal{Z}^N \mapsto \mathcal{W}$ be a randomized learning algorithm that maps a data set $S \in \mathcal{Z}^N$ to a model $A(S) \in \mathcal{W}$. Then $A$ is said to have $\gamma_{m,N}$-mean-uniform stability if for every $N \geq 1$,*

$$\sup_{S \doteq S'} \mathbb{E}_A \left[ \|A(S) - A(S')\| \right] \leq \gamma_{m,N}.$$

*Moreover, $A$ is said to have $\gamma_{m^2,N}$-mean-square-uniform stability if for every $N \geq 1$,*

$$\sup_{S \doteq S'} \mathbb{E}_A \left[ \|A(S) - A(S')\|^2 \right] \leq \gamma_{m^2,N}.$$

Here the algorithm outputs $A(S)$ and $A(S')$ share the same random bits associated with the algorithm. The above defined notion of mean-uniform stability is also known as Uniform Argument Stability (UAS) (Bassily et al., 2020), which was originally introduced by Liu et al. (2017) for non-parametric hypotheses. The notion of mean-square uniform stability is stronger than mean-uniform stability in the sense that the former naturally implies the latter such that $\gamma_{m,N} \leq \sqrt{\gamma_{m^2,N}}$. Moreover, we say a function $f$ is $G$-Lipschitz continuous over $\mathcal{W}$ if $|f(w) - f(w')| \leq G\|w - w'\|$ for all $w, w' \in \mathcal{W}$, and it is $L$-smooth if $\|\nabla f(w) - \nabla f(w')\| \leq L\|w - w'\|$ for all $w, w' \in \mathcal{W}$.

Inspired by a second moment bound for generalization error of uniformly stable algorithms from Bousquet et al. (2020, Section 5), we first establish the following lemma which states that if a randomized learning algorithm has $\gamma_{m,N}$-mean-uniform stability, then its on-average first moment generalization error bound will be as small as $\mathcal{O}(\gamma_{m,N} + \sqrt{\gamma_{m^2,N}} + \frac{1}{\sqrt{N}})$ when the loss function is Lipschitz continuous. This result is an adaptation of the second moment bound derived by Bousquet et al. (2020) to on-average uniform stable randomized algorithms, and that bound is also the source of $\gamma_{m^2,N}$ entering into play. For completeness, we provide a proof for this result in Appendix B.1.

**Lemma 1.** *Suppose that a randomized learning algorithm $A : \mathcal{Z}^N \mapsto \mathcal{W}$ has $\gamma_{m,N}$-mean-uniform stability and $\gamma_{m^2,N}$-mean-square-uniform stability. Assume that the loss function $\ell$ is $G$-Lipschitz with respect to its first argument and is bounded in the range of $[0, M]$. Then we have*

$$\mathbb{E}_{A,S} \left[ |R(A(S)) - R_S(A(S))| \right] \leq G\sqrt{\gamma_{m^2,N}} + G\gamma_{m,N} + \frac{M}{\sqrt{N}}.$$

**Remark 1.** *In comparison to the on-average bound $|\mathbb{E}_{A,S}[R(A(S)) - R_S(A(S))]| \leq G\gamma_{m,N}$ (see, e.g., Hardt et al., 2016, Theorem 2.2), our on-average first moment bound in Lemma 1 is substantially stronger and it turns out to play a crucial role in deriving high probability bounds for randomized algorithms. When $A$ is deterministic, the above bound reduces to the explicitly or implicitly known first moment bound for uniformly stable algorithms (see, e.g., Feldman & Vondrak, 2018; Bousquet et al., 2020) which is tighter in logarithmic factors than the one implied by the p-th moment inequality of (Bousquet et al., 2020, Theorem 4). In this case, our bound is stronger than the bound from Bousquet & Elisseeff (2002, Lemma 9) which essentially scales as $\frac{1}{\sqrt{N}} + \sqrt{\gamma_{m,N}}$ in our notation.*

### 2.2 MAIN RESULTS ON GENERALIZATION BOUND

Let us recall the subbagging process as described in Algorithm 1: we independently run $A$ over $K$ even and disjoint training subsets $\{S_k\}_{k \in [K]}$ to obtain solutions $\{A_k(S_k)\}_{k \in [K]}$, and then pick $A_{k^*}(S_{k^*})$ that has the smallest difference between training error and validation error (over the complementary training subset $S \setminus S_{k^*}$). The following theorem is our main result about the high probability generalization bound of the output $A_{k^*}(S_{k^*})$ evaluated over the entire training set $S$. See Appendix B.2 for its proof which builds largely on the first moment bound in Lemma 1 and the fact that at least one of the solutions generated by subbagging generalizes well with high probability.

**Theorem 1.** *Suppose that a randomized learning algorithm $A : \mathcal{Z}^N \mapsto \mathcal{W}$ has $\gamma_{m,N}$-mean-uniform stability and $\gamma_{m^2,N}$-mean-square-uniform stability as well. Assume that the loss function $\ell$ is G-Lipschitz with respect to its first argument and is bounded in the range of $[0, M]$. Then for any $\alpha, \delta \in (0, 1)$ and $K \geq \frac{1}{1-\alpha} \log(\frac{4}{\delta})$, with probability at least $1 - \delta$ over the randomness of $S$ and $\{A_k\}_{k\in[K]}$, the output of Algorithm 1 satisfies*

$$|R(A_{k^*}(S_{k^*})) - R_S(A_{k^*}(S_{k^*}))| \lesssim \frac{1}{\alpha K} \left( G\sqrt{\gamma_{m^2, \frac{N}{K}}} + G\gamma_{m, \frac{N}{K}} + M\sqrt{\frac{K}{N}} \right) + M\sqrt{\frac{\log(K/\delta)}{N}}.$$

**Remark 2.** *To gain some intuition on the superiority of our bound in Theorem 1, let us consider $K \asymp \log\left(\frac{1}{\delta}\right)$ as allowed in the conditions. If $\gamma_{m,N} \lesssim \frac{1}{\sqrt{N}}$ and $\gamma_{m^2,N} \lesssim \frac{1}{N}$, then our high-probability bound in Theorem 1 roughly scales as $\sqrt{\frac{\log(1/\delta)}{N}}$ which is sharper than the on-average bounds in equation 2 and equation 3 with $\gamma_N \lesssim \frac{1}{\sqrt{N}}$. More precise consequences of these general results on SGD and deterministic uniformly stable estimators such as ERMs and full gradient descent method will be discussed shortly in the sections to follow.*

**Remark 3.** *In sharp contrast to the bound in equation 5 that requires high probability uniform stability and the bound in equation 6 that assumes uniform stability over the random bits of algorithm, our bound in Theorem 1 holds under a substantially milder notion of mean(-square)-uniform stability over data. In terms of the tightness of bound, note that the confidence term $\sqrt{\frac{\log(1/\delta)}{N}}$ is necessary even for an algorithm with fixed output. The uniform stability terms $\gamma_{m, \frac{N}{K}}$ and $\gamma_{m^2, \frac{N}{K}}$ are also near-tight as the the algorithm output can change arbitrarily with respect to these quantities.*

## 2.3 ON EXCESS RISK BOUNDS

To understand the optimization performance of a randomized learning algorithm $A$ with confidence-boosting, we further study here the excess risk bounds of Algorithm 1 which are of special interest for stochastic convex optimization problems. In the following analysis, the global minimizer of the population risk and in-expectation empirical risk sub-optimality of the randomized algorithm are respectively denoted by

$$w^* := \arg\min_{w \in \mathcal{W}} R(w) \quad \text{and} \quad \Delta_{\text{opt}} := \mathbb{E}_{A,S}\left[ R_S(A(S)) - \min_{w \in \mathcal{W}} R_S(w) \right]. \tag{7}$$

In order to derive the excess risk guarantees, we first need to slightly modify the output of Algorithm 1 as $A_{k^*}(S_{k^*})$ where $k^* = \arg\min_{k \in [K]} R_{S \setminus S_k}(A_k(S_k))$. The following theorem is our main result about the high probability excess risk bounds of such a modified output of confidence-boosting. See Appendix B.3 for its proof.

**Theorem 2.** *Suppose that a randomized learning algorithm $A : \mathcal{Z}^N \mapsto \mathcal{W}$ has $\gamma_{m,N}$-mean-uniform stability. Assume that the loss function $\ell$ is G-Lipschitz with respect to its first argument and is bounded in the range of $[0, M]$. Then for any $\alpha, \delta \in (0, 1)$ and $K \geq \frac{1}{1-\alpha} \log(\frac{4}{\delta})$, with probability at least $1 - \delta$ over the randomness of $S$ and $\{A_k\}_{k\in[K]}$, the modified output of Algorithm 1 satisfies*

$$R(A_{k^*}(S_{k^*})) - R(w^*) \lesssim \frac{1}{\alpha} \left( G\gamma_{m, \frac{N}{K}} + \Delta_{opt} \right) + M\sqrt{\frac{\log(K/\delta)}{N}}.$$

**Remark 4.** *Unlike the generalization error bounds, the excess risk bounds established in Theorem 2 are not relying on the mean-square uniform stability of the algorithm, but with an additional term of in-expectation optimization error for minimizing the empirical risk. For deterministic optimization algorithms such as ERMs with $\Delta_{opt} = 0$, similar excess risk bounds can be implied by the generic results of Shalev-Shwartz et al. (2010, Theorem 26) developed for the confidence-boosting approach.*

**Remark 5.** *Consider $K \asymp \log\left(\frac{1}{\delta}\right)$ and $\gamma_{m,N} \lesssim \frac{1}{\sqrt{N}}$. Then the bound in Theorem 2 roughly scales as $\mathcal{O}(\sqrt{\frac{\log(1/\delta)}{N}} + \Delta_{opt})$.*

Finally, we comment on the difference between the generalization error and excess risk analysis inside the considered confidence-boosting framework. Since the excess risk is non-negative and

---

**Algorithm 2:** `SGD via With-Replacement Sampling` ($A_{\text{SGD-w}}$)

---

**Input** : Data set $S = \{z_i\}_{i \in [N]} \overset{\text{i.i.d.}}{\sim} \mathcal{D}^N$, step-sizes $\{\eta_t\}_{t \geq 1}$, #iterations $T$, initialization $w_0$.
**Output**: $\bar{w}_T = \frac{1}{T} \sum_{t \in [T]} w_t$.
**for** $t = 1, 2, ..., T$ **do**
     Uniformly randomly sample an index $\xi_t \in [N]$ with replacement;
     Compute $w_t = \Pi_{\mathcal{W}} (w_{t-1} - \eta_t \nabla_w \ell(w_{t-1}; z_{\xi_t}))$.
**end**

---

its in-expectation bound is standardly known for uniformly stable learning algorithms, the high-confidence bound in Theorem 2 can be easily derived via invoking Markov inequality to the independent runs of algorithm over $K$ disjoint subsets. The generalization error analysis, however, is way more challenging in the sense that establishing tight in-expectation first moment generalization bound (see Lemma 1) for uniformly stable randomized algorithms is by itself highly non-trivial.

## 3 IMPLICATIONS FOR STOCHASTIC GRADIENT DESCENT

In this section we demonstrate the applications of the generic bound in Theorem 1 to the widely used SGD algorithm. We focus on a variant of SGD under with-replacement sampling as outlined in Algorithm 2, which we call $A_{\text{SGD-w}}$. In what follows, we denote by $\{A_{\text{SGD-w},k}\}_{k \in K}$ the outputs of $A_{\text{SGD-w}}$ over subsets $\{S_k\}_{k \in K}$ when applied with Algorithm 1. Our results readily extend to the without-replacement variant of SGD and the corresponding results can be found in Appendix D.

### 3.1 CONVEX OPTIMIZATION WITH SMOOTH LOSS

For smooth and convex losses such as logistic loss, we can derive the following result as a direct consequence of Theorem 1 with $\alpha = 1/2$ to $A_{\text{SGD-w}}$. See Appendix C.1 for its proof.

**Corollary 1.** *Suppose that the loss function is $\ell(\cdot; \cdot)$ is convex, $G$-Lipschitz and $L$-smooth with respect to its first argument, and is bounded in the range of $[0, M]$. Consider Algorithm 1 specified to $A_{SGD-w}$ with learning rate $\eta_t \leq 2/L$ for all $t \geq 1$. Then for any $\delta \in (0, 1)$ and $K \asymp \log(\frac{4}{\delta})$, with probability at least $1 - \delta$ over the randomness of $S$ and $\{A_{SGD-w,k}\}_{k \in [K]}$, the generalization error is upper bounded as $|R(A_{SGD-w,k^*}(S_{k^*})) - R_S(A_{SGD-w,k^*}(S_{k^*}))| \lesssim$*

$$G^2 \sqrt{\frac{1}{N} \left( \sum_{t=1}^{T} \eta_t^2 + \frac{1}{N} \left( \sum_{t=1}^{T} \eta_t \right)^2 \right)} + \frac{G^2}{N} \sum_{t=1}^{T} \eta_t + M \sqrt{\frac{\log(1/\delta)}{N}}.$$

**Remark 6.** *For the conventional step-size choice of $\eta_t = \frac{2}{L\sqrt{t}}$, the high probability generalization bound in Corollary 1 is dominated by $\mathcal{O}(\sqrt{\frac{\log(T)}{N}} + \frac{\sqrt{T} + \sqrt{N \log(1/\delta)}}{N})$, which matches the corresponding $\mathcal{O}(\frac{\sqrt{T}}{N})$ in-expectation bound for SGD with smooth convex losses (Hardt et al., 2016).*

### 3.2 CONVEX OPTIMIZATION WITH NON-SMOOTH LOSS

Now we turn to study the case where the loss is convex but not necessarily smooth, such as the hinge loss and absolute loss. The following result as a direct consequence of Theorem 1 for the specification of Algorithm 1 to $A_{\text{SGD-w}}$ with non-smooth convex loss and time varying learning rate $\{\eta_t\}_{t \geq 1}$. Its proof is provided in Appendix C.2.

**Corollary 2.** *Suppose that the loss function is $\ell(\cdot; \cdot)$ is convex and $G$-Lipschitz with respect to its first argument, and is bounded in the range of $[0, M]$. Consider Algorithm 1 specified to $A_{SGD-w}$. Then for any $\delta \in (0, 1)$ and $K \asymp \log(\frac{4}{\delta})$, with probability at least $1 - \delta$ over the randomness of $S$ and $\{A_{SGD-w,k}\}_{k \in [K]}$, the generalization error satisfies $|R(A_{SGD-w,k^*}(S_{k^*})) - R_S(A_{SGD-w,k^*}(S_{k^*}))| \lesssim$*

$$G^2 \sqrt{\sum_{t=1}^{T} \eta_t^2 + \frac{1}{N^2} \left( \sum_{t=1}^{T} \eta_t \right)^2} + G^2 \sqrt{\sum_{t=1}^{T} \eta_t^2} + \frac{G^2}{N} \sum_{t=1}^{T} \eta_t + M \sqrt{\frac{\log(1/\delta)}{N}}.$$

**Remark 7.** *For constant rates $\eta_t \equiv \eta$, Corollary 2 admits a high probability generalization bound of scale $\mathcal{O}(\eta\sqrt{T} + \eta\frac{T}{N} + \sqrt{\frac{\log(1/\delta)}{N}})$ which matches the near-optimal rate by Bassily et al. (2020, Theorem 3.3). More importantly, our deviation bound in Corollary 2 still holds for time varying learning rates.*

### 3.3 NON-CONVEX OPTIMIZATION WITH SMOOTH LOSS

We further study the performance of Algorithm 1 for SGD on smooth but not necessarily convex loss functions, such as normalized sigmoid loss (Mason et al., 1999). The following result is a direct application of Theorem 1 to $A_{SGD-w}$ with smooth non-convex loss. See Appendix C.3 for its proof.

**Corollary 3.** *Suppose that the loss function is $\ell(\cdot;\cdot)$ is $G$-Lipschitz and $L$-smooth with respect to its first argument, and is bounded in the range of $[0,M]$. Consider Algorithm 1 specified to $A_{SGD-w}$ with $\eta_t \leq \frac{1}{L}$. Let $u_t := \eta_t^2 + 2\eta_t\sum_{\tau=1}^{t-1}\exp(L\sum_{i=\tau+1}^{t-1}\eta_i)\eta_\tau$ for all $t \geq 1$. Then for any $\delta \in (0,1)$ and $K \asymp \log(\frac{4}{\delta})$, with probability at least $1-\delta$ over the randomness of $S$ and $\{A_{SGD-w,k}\}_{k\in[K]}$, the generalization error is upper bounded as $|R(A_{SGD-w,k^*}(S_{k^*})) - R_S(A_{SGD-w,k^*}(S_{k^*}))| \lesssim$*

$$G^2\sqrt{\frac{1}{N}\sum_{t=1}^{T}\exp\left(L\sum_{\tau=t+1}^{T}\eta_\tau\right)u_t} + \frac{G^2}{N}\sum_{t=1}^{T}\exp\left(L\sum_{\tau=t+1}^{T}\eta_\tau\right)\eta_t + M\sqrt{\frac{\log(1/\delta)}{N}}.$$

**Remark 8.** *For the constant learning rates $\eta_t \equiv \frac{1}{LT}$, Corollary 3 admits high probability generalization bound of scale $\mathcal{O}(\sqrt{\frac{\log(1/\delta)}{N}})$. For time decaying learning rates $\eta_t = \frac{1}{L\nu t}$ with arbitrary $\nu \geq 1$, it can be verified that the corresponding bound is of scale $\mathcal{O}(\sqrt{\frac{T^{1/\nu}\log(T)}{\nu N}} + \sqrt{\frac{\log(1/\delta)}{N}})$.*

## 4 IMPLICATIONS FOR DETERMINISTIC UNIFORMLY STABLE ALGORITHMS

This section is devoted to showing that confidence-boosting is also beneficial for deriving stronger generalization bounds for uniformly stable deterministic learning algorithms. First we note that when there is no internal randomness in $A$, the definition of mean(-square)-uniform stability reduces to the conventional concept of $\gamma_N$-uniform stability for deterministic algorithms given by

$$\sup_{S \doteq S'} \|A(S) - A(S')\| \leq \gamma_N = \gamma_{m,N} = \sqrt{\gamma_{m^2,N}}.$$

Let us now consider a specification of Algorithm 1 to a uniformly stable deterministic algorithm $A$. Since there is no randomness contained in $A$, we have that $A_k(S_k) = A(S_k)$ for all $k \in [K]$ in such a deterministic case. Then the following result is a direct consequence of Theorem 1 when applied to the considered deterministic learning regime.

**Corollary 4.** *Suppose that a deterministic learning algorithm $A : \mathcal{Z}^N \mapsto \mathcal{W}$ has $\gamma_N$-uniform stability. Assume that the loss function $\ell$ is $G$-Lipschitz with respect to its first argument and is bounded in $[0,M]$. Then for any $\alpha,\delta \in (0,1)$ and $K \geq \frac{\log(4/\delta)}{1-\alpha}$, with probability at least $1-\delta$ over the randomness of $S$, the output of Algorithm 1 satisfies*

$$|R(A(S_{k^*})) - R_S(A(S_{k^*}))| \lesssim \frac{1}{\alpha K}\left(G\gamma_{\frac{N}{K}} + M\sqrt{\frac{K}{N}}\right) + M\sqrt{\frac{\log(K/\delta)}{N}}.$$

To demonstrate the superiority of our bound over prior ones, let us consider $\alpha = 0.5$ and $K \asymp \log(\frac{1}{\delta})$ in the above corollary. In the regime $\gamma_N \lesssim \frac{1}{\sqrt{N}}$ which is of interest in many popular deterministic learning paradigms/algorithms such as regularized ERM (Shalev-Shwartz et al., 2009) and full gradient descent (Feldman & Vondrak, 2019), Corollary 4 implies a generalization bound for $A(S_{k^*})$ over the data set $S$ that scales as $|R(A(S_{k^*})) - R_S(A(S_{k^*}))| \lesssim \sqrt{\frac{\log(1/\delta)}{N}}$. In comparison, the best known bound in equation 5 essentially from Bousquet et al. (2020) gives (keep in mind that $\delta' = 0$ in the deterministic case) $|R(A(S)) - R_S(A(S))| \lesssim \frac{\log(N)}{\sqrt{N}}\log(\frac{1}{\delta})$. As we can see that inside the carefully designed framework of confidence-bossting via subbagging, our generalization bound gets rid of the logarithmic factor $\log(N)$ from the above best known result, though the generalization is with respect to the estimation over a specific part of the sample. We expect this result will fuel future research towards fully resolving the corresponding open question raised in Bousquet et al. (2020).

## 5 OTHER RELATED WORK

The idea of using stability of a learning algorithm, namely the sensitivity of estimated model to the changes in training data, for generalization performance analysis dates back to the seventies (Vapnik & Chervonenkis, 1974; Rogers & Wagner, 1978; Devroye & Wagner, 1979). For deterministic learning algorithms, algorithmic stability has been extensively studied with a bunch of applications to establishing strong generalization and excess risk bounds for stable learning models like $k$-NN and regularized ERMs (Bousquet & Elisseeff, 2002; Zhang, 2003; Klochkov & Zhivotovskiy, 2021). The stability theory for randomized learning algorithms was formally introduced and investigated by Elisseeff et al. (2005). In a recent breakthrough work (Hardt et al., 2016), it was shown in that the solution obtained via stochastic gradient descent is expected to be stable and generalize well for smooth convex and non-convex loss functions. For non-smooth convex losses, the stability induced generalization bounds of SGD have been established in expectation (Lei & Ying, 2020) or deviation (Bassily et al., 2020). In Kuzborskij & Lampert (2018), a set of data-dependent generalization bounds for SGD were derived based on the stability of algorithm. More broadly, generalization bounds for stable learning algorithms that converge to global minima were established in Charles & Papailiopoulos (2018); Lei & Ying (2021). For non-convex sparse learning, algorithmic stability theory has been applied to derive the generalization bounds of the popularly used iterative hard thresholding (IHT) algorithm (Yuan & Li, 2021). The uniform stability bounds on SGD have also been extensively used for designing differential privacy stochastic optimization algorithms (Bassily et al., 2019; Feldman et al., 2020).

Bagging (or bootstrap aggregating) is one of the earliest and most popular ensemble methods that has been widely applied to reduce the variance for unstable learning algorithms such as decision tree and neural networks (Breiman, 1996; Opitz & Maclin, 1999), and sometimes stable algorithms such as SVMs (Valentini & Dietterich, 2003). As an important variant of bagging, subbagging has been proposed to reduce the computational cost of bagging via training base models under without-replacement sampling (Bühlmann, 2012). The stability and generalization bounds of bagging have been analyzed for both uniform (Elisseeff et al., 2005) and non-uniform (Foster et al., 2019) averaging schemes. Unlike these prior results for bagging with averaging aggregation, our bounds are obtained based on a confidence-boosting greedy aggregation scheme which turns out to yield sharper dependence on the uniform stability parameter.

The confidence-boosting technique has long been applied for obtaining sharp high-probability excess risk bounds from the corresponding strong in-expectation bounds (Shalev-Shwartz et al., 2010; Mehta, 2017). For generic statistical learning problems, confidence-boosting has been used to convert any low-confidence learning algorithm with linear dependence on $1/\delta$ to a high-confidence algorithm with logarithmic factor $\log(1/\delta)$. For learning with exp-concave losses, a relevant ERM estimator with in-expectation fast rate of convergence was converted to a high-confidence learning algorithm with an almost identical fast rate of convergence up to a logarithmic factor on $1/\delta$ (Mehta, 2017). While sharing a similar spirit of boosting the confidence, our generalization analysis is substantially more challenging than those prior excess risk analysis in terms of tightly deriving in-expectation first moment generalization bound for uniformly stable randomized algorithms.

## 6 CONCLUSIONS

In this paper we presented a generic confidence-boosting method for deriving near-optimal high probability generalization bounds for uniformly stable randomized learning algorithms. At a nutshell, our main results in Theorem 1 and Theorem 2 reveal that a carefully designed subbagging process in Algorithm 1 can yield high-confidence generalization and risk bounds under the notion of mean(-square)-uniform stability. Our theory has been substantialized to SGD on both convex and non-convex losses to obtain stronger generalization bounds especially in the case of time decaying learning rates. When reduced to deterministic algorithms, the proposed method removes a logarithmic factor on sample size from the best known bounds. While sharper in the dependence on sample size, our confidence-boosting results are only applicable to one of the independent runs of algorithm $A$ over $K$ disjoint training subsets of equal size with $K \asymp \log\left(\frac{1}{\delta}\right)$. It is so far not clear if these near-optimal bounds can be further extended to the full-batch setting where the generalization is with respect to the evaluation of algorithm over the entire sample. We leave the full understanding of such an open issue raised by Bousquet et al. (2020) for future investigation.

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

# A   AUXILIARY LEMMAS

We need the following lemma from Hardt et al. (2016) which shows that SGD iteration is non-expansive on convex and smooth loss.

**Lemma 2** (Hardt et al. (2016)). *Assume that $f$ is convex and $L$-smooth. Then for any $w, w' \in \mathcal{W}$ and $\alpha \leq 2/L$, we have the following bound holds*

$$\|w - \alpha \nabla f(w) - (w' - \alpha \nabla f(w'))\| \leq \|w - w'\|.$$

The following lemma, which can be proved by induction (see, e.g., Schmidt et al., 2011), will be used to prove the main results in Section 3.

**Lemma 3.** *Assume that the nonnegative sequence $\{u_\tau\}_{\tau \geq 1}$ satisfies the following recursion for all $t \geq 1$:*

$$u_t^2 \leq S_t + \sum_{\tau=1}^{t} \alpha_\tau u_\tau,$$

*with $\{S_\tau\}_{\tau \geq 1}$ an increasing sequence, $S_0 \geq u_0^2$ and $\alpha_\tau \geq 0$ for all $\tau$. Then, the following inequality holds for all $t \geq 1$:*

$$u_t \leq \sqrt{S_t} + \sum_{\tau=1}^{t} \alpha_\tau.$$

# B   PROOFS FOR THE RESULTS IN SECTION 2

In this section, we present the technical proofs for the main results stated in Section 2.

## B.1   PROOF OF LEMMA 1

We need the following lemma essentially from Bousquet et al. (2020) that provides a first moment bound for the sum of random functions.

**Lemma 4.** *Let $S = \{Z_1, Z_2, ..., Z_N\}$ be a set of i.i.d. random variables valued in $\mathcal{Z}$. Let $g_1, ..., g_N$ be a set of measurable functions $g_i : \mathcal{Z}^N \mapsto \mathbb{R}$ that satisfy $\mathbb{E}_S[g_i^2(S)] \leq M^2$ and $\mathbb{E}_{Z_i}[g_i(S)] = 0$ for all $i \in [N]$. Then we have*

$$\mathbb{E}_S\left[\left|\sum_{i=1}^{N} g_i(S)\right|\right] \leq \sqrt{\sum_{i \neq j} \mathbb{E}_{S,S^{(j)}}\left[\left(g_i(S) - g_i(S^{(j)})\right)^2\right]} + M\sqrt{N},$$

*where $S^{(j)} = \{Z_1, ..., Z_{j-1}, Z_j', Z_{j+1}, ..., Z_N\}$ and $S' = \{Z_1', Z_2', ..., Z_N'\}$ is another i.i.d. sample from the same distribution as that of $S$.*

*Proof.* We reproduce the proof in view of the argument in Bousquet et al. (2020, Section 5) showing that $\{g_i\}$ are weakly correlated. For any $i \neq j$, since $\mathbb{E}_{Z_i}[g_i(S)] = 0$ and $\mathbb{E}_{Z_j}[g_j(S)] = 0$, we can verify that

$$\mathbb{E}_S\left[g_i(S^{(j)})g_j(S)\right] = \mathbb{E}_{S \setminus Z_j}\left[\mathbb{E}_{Z_j}\left[g_i(S^{(j)})g_j(S) \mid S \setminus Z_j\right]\right] = \mathbb{E}_{S \setminus Z_j}\left[g_i(S^{(j)})\mathbb{E}_{Z_j}\left[g_j(S) \mid S \setminus Z_j\right]\right] = 0,$$

where we have used the independence of the elements in $S \cup \{Z_j'\}$. Similarly, we can show that

$$\mathbb{E}_S\left[g_i(S)g_j(S^{(i)})\right] = \mathbb{E}_S\left[g_i(S^{(j)})g_j(S^{(i)})\right] = 0.$$

Then it follows that for any $i \neq j$,

$$
\begin{aligned}
|\mathbb{E}_S[g_i(S)g_j(S)]| &= \left|\mathbb{E}_{S,S^{(i)},S^{(j)}}[g_i(S)g_j(S)]\right| \\
&= \left|\mathbb{E}_{S,S^{(i)},S^{(j)}}\left[(g_i(S) - g_i(S^{(j)}))(g_j(S) - g_j(S^{(i)}))\right]\right| \\
&\leq \mathbb{E}_{S,S^{(i)},S^{(j)}}\left[\left|(g_i(S) - g_i(S^{(j)}))(g_j(S) - g_j(S^{(i)}))\right|\right].
\end{aligned}
$$

Based on the above bound and Jensen's inequality we get

$$
\mathbb{E}_S \left[ \left| \sum_{i=1}^N g_i(S) \right| \right]
$$

$$
\leq \sqrt{ \mathbb{E}_S \left[ \left( \sum_{i=1}^N g_i(S) \right)^2 \right] } = \sqrt{ \sum_{i \neq j} \mathbb{E}_S \left[ g_i(S) g_j(S) \right] + \sum_{i=1}^N \mathbb{E}_S[g_i^2(S)] }
$$

$$
\leq \sqrt{ \sum_{i \neq j} \mathbb{E}_{S,S^{(i)},S^{(j)}} \left[ |(g_i(S) - g_i(S^{(j)}))(g_j(S) - g_j(S^{(i)}))| \right] } + M\sqrt{N}
$$

$$
\leq \sqrt{ \frac{1}{2} \sum_{i \neq j} \mathbb{E}_{S,S^{(i)},S^{(j)}} \left[ \left( g_i(S) - g_i(S^{(j)}) \right)^2 + \left( g_j(S) - g_j(S^{(i)}) \right)^2 \right] } + M\sqrt{N}
$$

$$
= \sqrt{ \sum_{i \neq j} \mathbb{E}_{S,S^{(j)}} \left[ \left( g_i(S) - g_i(S^{(j)}) \right)^2 \right] } + M\sqrt{N}.
$$

This proves the desired bound. $\qquad\square$

Now we are ready to prove the result in Lemma 1.

*Proof of Lemma 1.* Let us consider $h_i(S) := R(A(S)) - \ell(A(S); Z_i)$ and $g_i(S) = h_i(S) - \mathbb{E}_{Z_i}[h_i(S)]$ for $i \in [N]$. Then by assumption we have

$$
\mathbb{E}_{Z_i}[g_i(S)] = 0, \quad \mathbb{E}_S[g_i^2(S)] \leq M^2.
$$

For each $i \in [N]$, let $S^{(i)}$ denote a random data set that is identical to $S$ except that one of the $Z_i$ is replaced by another random sample $Z_i'$. For any $i \neq j$, since the loss is non-negative and $G$-Lipschitz, it can be verified that

$$
|g_i(S) - g_i(S^{(j)})| \leq \max \left\{ \left| h_i(S) - h_i(S^{(j)}) \right|, \left| \mathbb{E}_{Z_i}[h_i(S) - h_i(S^{(j)})] \right| \right\}
$$

$$
\leq \max \left\{ G\|A(S) - A(S^{(j)})\|, \mathbb{E}_{Z_i} \left[ G\|A(S) - A(S^{(j)})\| \right] \right\},
$$

which readily implies

$$
\mathbb{E}_{S,S^{(j)}} \left[ \left( g_i(S) - g_i(S^{(j)}) \right)^2 \right] \leq \mathbb{E}_{S,S^{(j)}} \left[ G^2 \|A(S) - A(S^{(j)})\|^2 \right].
$$

Then invoking Lemma 4 to $\{g_i\}$ yields

$$
\mathbb{E}_S \left[ \left| \sum_{i=1}^N g_i(S) \right| \right] \leq \sqrt{ \sum_{i \neq j} \mathbb{E}_{S,S^{(j)}} \left[ \left( g_i(S) - g_i(S^{(j)}) \right)^2 \right] } + M\sqrt{N}
$$

$$
\leq G \sqrt{ \sum_{i \neq j} \mathbb{E}_{S,S^{(j)}} \left[ \|A(S) - A(S^{(j)})\|^2 \right] } + M\sqrt{N}. \tag{A.1}
$$

Further, it can be verified that

$$
\mathbb{E}_S \left[ \left| \sum_{i=1}^N \mathbb{E}_{Z_i}[h_i(S)] \right| \right]
$$

$$
= \mathbb{E}_S \left[ \left| \sum_{i=1}^N \mathbb{E}_{Z_i}[R(A(S)) - \ell(A(S); Z_i)] \right| \right]
$$

$$
= \mathbb{E}_S \left[ \left| \sum_{i=1}^N \mathbb{E}_{Z_i}[\mathbb{E}_{Z_i'}[\ell(A(S); Z_i')] - \ell(A(S); Z_i)] \right| \right] \tag{A.2}
$$

$$
= \mathbb{E}_S \left[ \left| \sum_{i=1}^N \mathbb{E}_{Z_i} \left[ \mathbb{E}_{Z_i'}[\ell(A(S); Z_i')] - \mathbb{E}_{Z_i'}[\ell(A(S^{(i)}); Z_i')] \right] \right| \right]
$$

$$
\leq \mathbb{E}_{S,S^{(i)}} \left[ G\|A(S) - A(S^{(i)})\| \right].
$$

By combining equation A.1 and equation A.2 we obtain

$$
\mathbb{E}_{A,S}\left[|R(A(S)) - R_S(A(S))|\right]
$$

$$
= \frac{1}{N}\mathbb{E}_{A,S}\left[\left|\sum_{i=1}^{N}\left(g_i(S) + \mathbb{E}_{Z_i}[h_i(S)]\right)\right|\right]
$$

$$
\leq \frac{G}{N}\mathbb{E}_A\left[\sqrt{\sum_{i\neq j}\mathbb{E}_{S,S^{(j)}}\left[\|A(S) - A(S^{(j)})\|^2\right]}\right] + \frac{G}{N}\mathbb{E}_{A,S,S^{(i)}}\left[\|A(S) - A(S^{(i)})\|\right] + \frac{M}{\sqrt{N}}
$$

$$
\leq \frac{G}{N}\sqrt{\sum_{i\neq j}\mathbb{E}_{A,S,S^{(j)}}\left[\|A(S) - A(S^{(j)})\|^2\right]} + \frac{G}{N}\mathbb{E}_{A,S,S^{(i)}}\left[\|A(S) - A(S^{(i)})\|\right] + \frac{M}{\sqrt{N}}
$$

$$
\leq G\sqrt{\gamma_{\mathrm{m}^2,N}} + G\gamma_{\mathrm{m},N} + \frac{M}{\sqrt{N}},
$$

where in the last inequality we have used the stability conditions on $A$. The proof is completed. $\quad\square$

### B.2 Proof of Theorem 1

We first establish the following intermediate result that captures the effects of subbagging on randomized algorithms: it basically tells that with $K \asymp \log(\frac{1}{\delta})$, at least one of the solutions generated by subbagging generalizes well with high probability.

**Lemma 5.** *Suppose that a randomized learning algorithm $A : \mathcal{Z}^N \mapsto \mathcal{W}$ has $\gamma_{m,N}$-mean-uniform stability and $\gamma_{m^2,N}$-mean-square-uniform stability as well. Assume that the loss function $\ell$ is $G$-Lipschitz with respect to its first argument and is bounded in the range of $[0, M]$. Then for any $\alpha, \delta \in (0,1)$ and $K \geq \frac{\log(2/\delta)}{1-\alpha}$, with probability at least $1 - \delta$ over the randomness of $\{(A_k, S_k)\}_{k\in[K]}$, the sequence $\{A_k(S_k)\}_{k\in[K]}$ generated by Algorithm 1 satisfies*

$$
\min_{k\in[K]}|R(A_k(S_k)) - R_{S_k}(A_k(S_k))| \lesssim \frac{1}{\alpha}\left(G\sqrt{\gamma_{m^2,\frac{N}{K}}} + G\gamma_{m,\frac{N}{K}} + M\sqrt{\frac{K}{N}}\right).
$$

*Proof.* From Lemma 1 we have that over randomized algorithm $A$ and data $\tilde{S}$ with $|\tilde{S}| = \frac{N}{K}$,

$$
\mathbb{E}_{A,\tilde{S}}\left[|R(A(\tilde{S})) - R_{\tilde{S}}(A(\tilde{S}))|\right] \leq G\sqrt{\gamma_{m^2,\frac{N}{K}}} + G\gamma_{m,\frac{N}{K}} + M\sqrt{\frac{K}{N}}.
$$

Since $\{A_k, S_k\}_{k\in[K]}$ are independent to each other, by Markov inequality we know that

$$
\mathbb{P}_{\{A_k,S_k\}}\left(\min_{k\in[K]}|R(A_k(S_k)) - R_{S_k}(A_k(S_k))| \geq \frac{1}{\alpha}\left(G\sqrt{\gamma_{m^2,\frac{N}{K}}} + G\gamma_{m,\frac{N}{K}} + M\sqrt{\frac{K}{N}}\right)\right) \leq \alpha^K \leq \delta,
$$

which implies the desired bound. $\quad\square$

Next we proceed to prove the main result in Theorem 1.

*Proof of Theorem 1.* Let us consider the following three events:

$$
\mathcal{E} := \left\{|R(A_{k^*}(S_{k^*})) - R_S(A_{k^*}(S_{k^*}))| \lesssim \frac{1}{\alpha K}\left(G\sqrt{\gamma_{m^2,\frac{N}{K}}} + G\gamma_{m,\frac{N}{K}} + M\sqrt{\frac{K}{N}}\right) + M\sqrt{\frac{K\log(K/\delta)}{(K-1)N}}\right\},
$$

$$
\mathcal{E}_1 := \left\{\max_{k\in[K]}|R(A_k(S_k)) - R_{S\setminus S_k}(A_k(S_k))| \lesssim M\sqrt{\frac{K\log(K/\delta)}{(K-1)N}}\right\},
$$

$$
\mathcal{E}_2 := \left\{\min_{k\in[K]}|R(A_k(S_k)) - R_{S_k}(A_k(S_k))| \lesssim \frac{1}{\alpha}\left(G\sqrt{\gamma_{m^2,\frac{N}{K}}} + G\gamma_{m,\frac{N}{K}} + M\sqrt{\frac{K}{N}}\right)\right\}.
$$

We can show that $\mathcal{E} \supseteq \mathcal{E}_1 \cap \mathcal{E}_2$. Indeed, suppose that $\mathcal{E}_1$ and $\mathcal{E}_2$ simultaneously occur. Consequently the following inequality is valid:

$$
\begin{aligned}
&\left|R(A_{k^*}(S_{k^*})) - R_S(A_{k^*}(S_{k^*}))\right| \\
=& \left|R(A_{k^*}(S_{k^*})) - \frac{1}{K} R_{S_{k^*}}(A_{k^*}(S_{k^*})) - \frac{K-1}{K} R_{S \setminus S_{k^*}}(A_{k^*}(S_{k^*}))\right| \\
\leq& \frac{1}{K}\left|R(A_{k^*}(S_{k^*})) - R_{S_{k^*}}(A_{k^*}(S_{k^*}))\right| + \frac{K-1}{K}\left|R(A_{k^*}(S_{k^*})) - R_{S \setminus S_{k^*}}(A_{k^*}(S_{k^*}))\right| \\
\leq& \frac{1}{K}\left|R_{S \setminus S_{k^*}}(A_{k^*}(S_{k^*})) - R_{S_{k^*}}(A_{k^*}(S_{k^*}))\right| + \left|R(A_{k^*}(S_{k^*})) - R_{S \setminus S_{k^*}}(A_{k^*}(S_{k^*}))\right| \\
\overset{\zeta_1}{=}& \frac{1}{K} \min_{k \in [K]}\left|R_{S \setminus S_k}(A_k(S_k)) - R_{S_k}(A_k(S_k))\right| + \left|R(A_{k^*}(S_{k^*})) - R_{S \setminus S_{k^*}}(A_{k^*}(S_{k^*}))\right| \\
=& \frac{1}{K} \min_{k \in [K]}\left|R_{S \setminus S_k}(A_k(S_k)) - R(A_k(S_k)) + R(A_k(S_k)) - R_{S_k}(A_k(S_k))\right| \\
&+ \left|R(A_{k^*}(S_{k^*})) - R_{S \setminus S_{k^*}}(A_{k^*}(S_{k^*}))\right| \\
\leq& \frac{1}{K} \min_{k \in [K]}\left|R(A_k(S_k)) - R_{S_k}(A_k(S_k))\right| + \frac{1}{K} \max_{k \in [K]}\left|R_{S \setminus S_k}(A_k(S_k)) - R(A_k(S_k))\right| \\
&+ \left|R(A_{k^*}(S_{k^*})) - R_{S \setminus S_{k^*}}(A_{k^*}(S_{k^*}))\right| \\
\leq& \frac{1}{K} \min_{k \in [K]}\left|R(A_k(S_k)) - R_{S_k}(A_k(S_k))\right| + \frac{K+1}{K} \max_{k \in [K]}\left|R(A_k(S_k)) - R_{S \setminus S_k}(A_k(S_k))\right| \\
\overset{\zeta_2}{\lesssim}& \frac{1}{\alpha K}\left(G\sqrt{\gamma_{\mathrm{m}^2, \frac{N}{K}}} + G\gamma_{\mathrm{m}, \frac{N}{K}} + M\sqrt{\frac{K}{N}}\right) + M\sqrt{\frac{K \log(K/\delta)}{(K-1)N}},
\end{aligned}
$$

where in "$\zeta_1$" we have used the definition of $k^*$, and "$\zeta_2$" follows from $\mathcal{E}_1, \mathcal{E}_2$. With leading terms preserved in the above we can see that $\mathcal{E}$ occurs.

Next we can show that $\mathbb{P}_{S, \{A_k\}}(\overline{\mathcal{E}_1}) \leq \frac{\delta}{2}$. Toward this end, let us consider the following events for all $k \in [K]$:

$$
\mathcal{E}_1^k := \left\{\left|R(A_k(S_k)) - R_{S \setminus S_k}(A_k(S_k))\right| \lesssim M\sqrt{\frac{K \log(K/\delta)}{(K-1)N}}\right\}.
$$

Clearly, it is true that $\mathcal{E}_1 = \bigcap_{k=1}^{K} \mathcal{E}_1^k$. It is sufficient to prove that $\mathbb{P}_{S, A_k}\left(\overline{\mathcal{E}_1^k}\right) \leq \frac{\delta}{2K}$ holds for each $k \in [K]$. Indeed, consider the random indication function $\beta(S, A_k) := \mathbf{1}_{\overline{\mathcal{E}_1^k}}$ associated with the event $\overline{\mathcal{E}_1^k}$. Then we have the following holds for each $k \in [K]$:

$$
\begin{aligned}
&\mathbb{P}_{S, A_k}\left(\overline{\mathcal{E}_1^k}\right) \\
=& \mathbb{E}_{S, A_k}[\beta(S, A_k)] \\
=& \mathbb{E}_{A_k, S_k}\left[\mathbb{E}_{S \setminus S_k | A_k, S_k}[\beta(S, A_k) \mid A_k, S_k]\right] \\
\overset{\zeta_1}{=}& \mathbb{E}_{A_k, S_k}\left[\mathbb{E}_{S \setminus S_k}[\beta(S, A_k) \mid A_k, S_k]\right] \\
=& \mathbb{E}_{A_k, S_k}\left[\mathbb{P}_{S \setminus S_k}\left(\left|R(A_k(S_k)) - R_{S \setminus S_k}(A_k(S_k))\right| \gtrsim M\sqrt{\frac{K \log(K/\delta)}{(K-1)N}}\right) \mid A_k, S_k\right] \\
\overset{\zeta_2}{\leq}& \mathbb{E}_{A_k, S_k}\left[\frac{\delta}{2K} \mid A_k, S_k\right] = \frac{\delta}{2K},
\end{aligned}
$$

where in "$\zeta_1$" we have used the independence between $\{A_k, S_k\}$ and $S \setminus S_k$, and "$\zeta_2$" is due to Hoeffding's inequality conditioned on $\{A_k, S_k\}$, keeping in mind that $A_k(S_k)$ is independent on the data set $S \setminus S_k$ of size $(1 - 1/K)N$. It follows by union probability that

$$
\mathbb{P}_{S, \{A_k\}}(\overline{\mathcal{E}_1}) = \mathbb{P}_{S, \{A_k\}}\left(\bigcup_{k=1}^{K} \overline{\mathcal{E}_1^k}\right) \leq \sum_{k=1}^{K} \mathbb{P}_{S, A_k}\left(\overline{\mathcal{E}_1^k}\right) \leq \frac{\delta}{2}.
$$

Further, from the part(b) of Lemma 5 we have $\mathbb{P}_{S,\{A_k\}}(\bar{\mathcal{E}}_2) \leq \frac{\delta}{2}$. Combining this and the preceding bound yields

$$\mathbb{P}_{S,\{A_k\}}(\mathcal{E}) \geq \mathbb{P}_{S,\{A_k\}}(\mathcal{E}_1 \cap \mathcal{E}_2) \geq 1 - \mathbb{P}_{S,\{A_k\}}(\bar{\mathcal{E}}_1) - \mathbb{P}_{S,\{A_k\}}(\bar{\mathcal{E}}_2) \geq 1 - \frac{\delta}{2} - \frac{\delta}{2} = 1 - \delta.$$

This implies the desired result in part(b) as $K/(K-1) \leq 2$ for $K \geq 2$. $\qquad\square$

### B.3 PROOF OF THEOREM 2

We first present the following simple lemma about the in-expectation risk bounds of a randomized algorithm which will be used in our analysis.

**Lemma 6.** *Suppose that a randomized learning algorithm* $A : \mathcal{Z}^N \mapsto \mathcal{W}$ *has* $\gamma_{m,N}$*-mean-uniform stability. Assume that the loss function* $\ell$ *is* $G$*-Lipschitz with respect to its first argument. Then we have*

$$\mathbb{E}_{A,S}[R(A(S)) - R(w^*)] \leq G\gamma_{m,N} + \Delta_{opt}.$$

*Proof.* By risk decomposition we can show that

$$
\begin{aligned}
&\mathbb{E}_{A,S}[R(A(S)) - R(w^*)] \\
=&\mathbb{E}_{A,S}[R(A(S)) - R_S(A(S)) + R_S(A(S)) - R_S(w^*) + R_S(w^*) - R(w^*)] \\
\leq&|\mathbb{E}_{A,S}[R(A(S)) - R_S(A(S))]| + \Delta_{\text{opt}} \\
\leq&G\gamma_{\text{m},N} + \Delta_{\text{opt}},
\end{aligned}
$$

where in the last inequality we have used the on-average generalization bound by Hardt et al. (2016, Theorem 2.2). $\qquad\square$

**Lemma 7.** *Suppose that a randomized learning algorithm* $A : \mathcal{Z}^N \mapsto \mathcal{W}$ *has* $\gamma_{m,N}$*-mean-uniform stability. Assume that the loss function* $\ell$ *is* $G$*-Lipschitz with respect to its first argument. Then for any* $\alpha, \delta \in (0,1)$ *and* $K \geq \frac{\log(2/\delta)}{1-\alpha}$, *with probability at least* $1 - \delta$ *over the randomness of* $\{(A_k, S_k)\}_{k \in [K]}$, *the sequence* $\{A_k(S_k)\}_{k \in [K]}$ *generated by Algorithm 1 with the modified output satisfies*

$$\min_{k \in [K]} R(A_k(S_k)) - R(w^*) \lesssim \frac{1}{\alpha}\left(G\gamma_{m,\frac{N}{K}} + \Delta_{opt}\right).$$

*Proof.* Recall the modified output $A_{k^*}(S_{k^*})$ where $k^* = \arg\min_{k \in [K]} R_{S \setminus S_k}(A_k(S_k))$. From Lemma 6 we have that over randomized algorithm $A$ and data $\tilde{S}$ with $|\tilde{S}| = \frac{N}{K}$,

$$\mathbb{E}_{A,\tilde{S}}\left[R(A(\tilde{S})) - R(w^*)\right] \leq G\gamma_{m,\frac{N}{K}} + \Delta_{\text{opt}}.$$

Since $\{A_k, S_k\}_{k \in [K]}$ are independent to each other, by Markov inequality we know that

$$\mathbb{P}_{\{A_k, S_k\}}\left(\min_{k \in [K]} R(A_k(S_k)) - R(w^*) \geq \frac{1}{\alpha}\left(G\gamma_{m,\frac{N}{K}} + \Delta_{\text{opt}}\right)\right) \leq \alpha^K \leq \delta,$$

which implies the desired bound in part(b). $\qquad\square$

Next we proceed to prove the main result in Theorem 2.

*Proof of Theorem 2.* Let us consider the following three events:

$$\mathcal{E} := \left\{|R(A_{k^*}(S_{k^*})) - R_S(A_{k^*}(S_{k^*}))| \lesssim \frac{1}{\alpha}\left(G\gamma_{m,\frac{N}{K}} + \Delta_{\text{opt}}\right) + M\sqrt{\frac{K\log(K/\delta)}{(K-1)N}}\right\},$$

$$\mathcal{E}_1 := \left\{\max_{k \in [K]} |R(A_k(S_k)) - R_{S \setminus S_k}(A_k(S_k))| \lesssim M\sqrt{\frac{K\log(K/\delta)}{(K-1)N}}\right\},$$

$$\mathcal{E}_2 := \left\{\min_{k \in [K]} R(A_k(S_k)) - R(w^*) \lesssim \frac{1}{\alpha}\left(G\gamma_{m,\frac{N}{K}} + \Delta_{\text{opt}}\right)\right\}.$$

Similarly, we show that $\mathcal{E} \supseteq \mathcal{E}_1 \cap \mathcal{E}_2$. Indeed, suppose that $\mathcal{E}_1$ and $\mathcal{E}_2$ simultaneously occur. Consequently the following inequality is valid:

$$
\begin{aligned}
& R(A_{k^*}(S_{k^*})) - R(w^*) \\
=& R(A_{k^*}(S_{k^*})) - R_{S \setminus S_{k^*}}(A_{k^*}(S_{k^*})) + R_{S \setminus S_{k^*}}(A_{k^*}(S_{k^*})) - R(w^*) \\
\overset{\zeta_1}{=}& R(A_{k^*}(S_{k^*})) - R_{S \setminus S_{k^*}}(A_{k^*}(S_{k^*})) + \min_{k \in [K]} R_{S \setminus S_k}(A_k(S_k)) - R(w^*) \\
=& R(A_{k^*}(S_{k^*})) - R_{S \setminus S_{k^*}}(A_{k^*}(S_{k^*})) + \min_{k \in [K]} \left\{ R_{S \setminus S_k}(A_k(S_k)) - R(A_k(S_k)) + R(A_k(S_k)) - R(w^*) \right\} \\
\leq& \min_{k \in [K]} (R(A_k(S_k)) - R(w^*)) + 2 \max_{k \in [K]} \left| R_{S \setminus S_k}(A_k(S_k)) - R(A_k(S_k)) \right| \\
\overset{\zeta_2}{\lesssim}& \frac{1}{\alpha} \left( G \gamma_{\mathrm{m}, \frac{N}{K}} + \Delta_{\mathrm{opt}} \right) + M \sqrt{\frac{K \log(K/\delta)}{(K-1)N}},
\end{aligned}
$$

where in "$\zeta_1$" we have used the definition of $k^*$, and "$\zeta_2$" follows from $\mathcal{E}_1, \mathcal{E}_2$. With leading terms preserved in the above we can see that $\mathcal{E}$ occurs.

Based on the same proof argument as that of the part(b) of Theorem 1 we can show that $\mathbb{P}_{S, \{A_k\}}(\bar{\mathcal{E}}_1) \leq \frac{\delta}{2}$. Further, from the part(b) of Lemma 7 we have $\mathbb{P}_{S, \{A_k\}}(\bar{\mathcal{E}}_2) \leq \frac{\delta}{2}$. Combining this and the preceding bound yields

$$
\mathbb{P}_{S, \{A_k\}}(\mathcal{E}) \geq \mathbb{P}_{S, \{A_k\}}(\mathcal{E}_1 \cap \mathcal{E}_2) \geq 1 - \mathbb{P}_{S, \{A_k\}}(\bar{\mathcal{E}}_1) - \mathbb{P}_{S, \{A_k\}}(\bar{\mathcal{E}}_2) \geq 1 - \frac{\delta}{2} - \frac{\delta}{2} = 1 - \delta.
$$

This implies the desired result in part(b) as $K/(K-1) \leq 2$ for $K \geq 2$. $\qquad \square$

## C    PROOFS FOR THE RESULTS IN SECTION 3

In this section, we present the technical proofs for the main results stated in Section 3.

### C.1    PROOF OF COROLLARY 1

We begin with presenting and proving the following lemma that gives the mean(-square)-uniform stability bounds for $A_{\mathrm{SGD-w}}$ on convex and smooth loss functions such as logistic loss.

**Lemma 8.** *Suppose that the loss function is $\ell(\cdot; \cdot)$ is convex, G-Lipschitz and L-smooth with respect to its first argument. Assume that $\eta_t \leq 2/L$ for all $t \geq 1$. Then $A_{\mathrm{SGD-w}}$ has mean-uniform stability such that*

$$
\sup_{S \doteq S'} \mathbb{E}_{A_{\mathrm{SGD-w}}} [\|A_{\mathrm{SGD-w}}(S) - A_{\mathrm{SGD-w}}(S')\|] \leq \frac{2G}{N} \sum_{t \in [T]} \eta_t,
$$

*and has mean-square-uniform stability such that*

$$
\sup_{S \doteq S'} \mathbb{E}_{A_{\mathrm{SGD-w}}} [\|A_{\mathrm{SGD-w}}(S) - A_{\mathrm{SGD-w}}(S')\|^2] \leq \frac{40G^2}{N} \left( \sum_{t=1}^{T} \eta_t^2 + \frac{1}{N} \left( \sum_{t=1}^{T} \eta_t \right)^2 \right).
$$

*Proof.* The first mean-uniform stability bound can be straightforwardly derived based on the argument of Hardt et al. (2016, Theorem 3.7). We focus on proving the second mean-square-uniform stability bound. For any pair of $S, S'$, let us define the sequences $\{w_t\}_{t \in [T]}$ and $\{w_t'\}_{t \in [T]}$ that are respectively generated over $S$ and $S'$ via $A_{\mathrm{SGD-w}}$ via sample path $\xi = \{\xi_t\}_{t \in [T]}$. Note by assumption that $w_0 = w_0'$. We distinguish the following two complementary cases.

**Case I:** $z_{\xi_t} = z'_{\xi_t}$. In this case, by invoking Lemma 2 we immediately get

$$
\begin{aligned}
\|w_t - w_t'\|^2 =& \|\Pi_{\mathcal{W}}(w_{t-1} - \eta_t \nabla_w \ell(w_{t-1}; z_{\xi_t})) - \Pi_{\mathcal{W}}(w_{t-1}' - \eta_t \nabla_w \ell(w_{t-1}'; z'_{\xi_t}))\|^2 \\
\leq& \|w_{t-1} - \eta_t \nabla_w \ell(w_{t-1}; z_{\xi_t}) - (w_{t-1}' - \eta_t \nabla_w \ell(w_{t-1}'; z'_{\xi_t}))\|^2 \qquad\qquad (A.3) \\
\leq& \|w_{t-1} - w_{t-1}'\|^2.
\end{aligned}
$$

**Case II:** $z_{\xi_t} \neq z'_{\xi_t}$. In this case, we have

$$
\begin{aligned}
\|w_t - w'_t\|^2 &= \|\Pi_{\mathcal{W}}(w_{t-1} - \eta_t \nabla f(w)) - \Pi_{\mathcal{W}}(w' - \alpha \nabla f(w'))\|^2 \\
&\leq \|w_{t-1} - \eta_t \nabla_w \ell(w_{t-1}; z_{\xi_t}) - (w'_{t-1} - \eta_t \nabla_w \ell(w'_{t-1}; z'_{\xi_t}))\|^2 \\
&\leq \left(\|w_{t-1} - w'_{t-1}\| + \eta_t(\|\nabla_w \ell(w_{t-1}; z_{\xi_t})\| + \|\nabla_w \ell(w'_{t-1}; z'_{\xi_t})\|)\right)^2 \quad \text{(A.4)} \\
&\leq \left(\|w_{t-1} - w'_{t-1}\| + 2G\eta_t\right)^2 \\
&= \|w_{t-1} - w'_{t-1}\|^2 + 4G\eta_t \|w_{t-1} - w'_{t-1}\| + 4G^2 \eta_t^2,
\end{aligned}
$$

where in the last but inequality we have used $\ell(\cdot; \cdot)$ is $G$-Lipschitz with respect to its first argument.

Let $\beta_t = \beta_t(S, S', \xi) := \mathbf{1}_{\left\{z_{\xi_t} \neq z'_{\xi_t}\right\}}$ be the random indication function associated with event $z_{\xi_t} \neq z'_{\xi_t}$. Based on the recursion forms equation A.3 and equation A.4 and the condition $w_0 = w'_0$ we can show that for all $t \geq 1$,

$$
\|w_t - w'_t\|^2 \leq \sum_{\tau=1}^t 4G\beta_\tau \eta_\tau \|w_{\tau-1} - w'_{\tau-1}\| + \sum_{\tau=1}^t 4G^2 \beta_\tau \eta_\tau^2.
$$

Then applying Lemma 3 with simple algebraic manipulation yields

$$
\|w_t - w'_t\|^2 \leq 8G^2 \left(\sum_{\tau=1}^t \beta_\tau \eta_\tau^2 + 4\left(\sum_{\tau=1}^t \beta_\tau \eta_\tau\right)^2\right).
$$

Since by assumption $S$ and $S'$ differ only in a single element, under the scheme of uniform sampling without replacement, we can see that $\beta_t(S, S', \xi) \sim \texttt{Bernoulli}(1/N)$ and $\{\beta_t(S, S', \xi)\}$ is an i.i.d. sequence of Bernoulli random variables. It follows that

$$
\begin{aligned}
&\mathbb{E}_{\xi_{[t]}} \left[\|w_t - w'_t\|^2\right] \\
\leq &8G^2 \left(\sum_{\tau=1}^t \mathbb{E}_{\xi_{[t]}}[\beta_\tau] \eta_\tau^2 + 4\mathbb{E}_{\xi_{[t]}} \left[\left(\sum_{\tau=1}^t \beta_\tau \eta_\tau\right)^2\right]\right) \\
= &8G^2 \left(\sum_{\tau=1}^t \mathbb{E}_{\xi_{[t]}}[\beta_\tau + 4\beta_\tau^2] \eta_\tau^2 + 4\sum_{\tau \neq \tau'} \mathbf{1}\mathbb{E}_{\xi_{[t]}}[\beta_\tau \beta_{\tau'}] \eta_\tau \eta_{\tau'}\right) \\
= &8G^2 \left(\frac{5}{N} \sum_{\tau=1}^t \eta_\tau^2 + \frac{4}{N^2} \left(\sum_{\tau=1}^t \eta_\tau\right)^2\right) \leq 40G^2 \left(\frac{1}{N} \sum_{\tau=1}^T \eta_\tau^2 + \frac{1}{N^2} \left(\sum_{\tau=1}^T \eta_\tau\right)^2\right),
\end{aligned}
$$

where we have used $\mathbb{E}_{\xi_t}[\beta_t] = \mathbb{E}_{\xi_t}[\beta_t^2] = \frac{1}{N}$. The convexity of squared Euclidean norm leads to

$$
\mathbb{E}_\xi \left[\|\bar{w}_T - \bar{w}'_T\|^2\right] \leq \frac{\sum_{t=1}^T \mathbb{E}_{\xi_{[t]}} \left[\|w_t - w'_t\|^2\right]}{T} \leq 40G^2 \left(\frac{1}{N} \sum_{t=1}^T \eta_t^2 + \frac{1}{N^2} \left(\sum_{t=1}^T \eta_t\right)^2\right).
$$

Note that the above holds for any $S \doteq S'$, i.e.,

$$
\sup_{S \doteq S'} \mathbb{E}_\xi \left[\|\bar{w}_T - \bar{w}'_T\|^2\right] \leq \frac{40G^2}{N} \left(\sum_{t=1}^T \eta_t^2 + \frac{1}{N} \left(\sum_{t=1}^T \eta_t\right)^2\right).
$$

The proof is concluded. $\qquad\square$

With Lemma 8 in place, we are ready to prove Corollary 1.

*Proof of Corollary 1.* From Lemma 8 we know that $A_{\text{SGD-w}}$ has mean-uniform stability with parameter

$$
\gamma_{\text{m},N} = \frac{2G}{N} \sum_{t \in [T]} \eta_t,
$$

and mean-square-uniform stability with parameter

$$
\gamma_{\mathrm{m}^2,N} = \frac{40G^2}{N}\left(\sum_{t=1}^{T}\eta_t^2 + \frac{1}{N}\left(\sum_{t=1}^{T}\eta_t\right)^2\right).
$$

The desired results then follow immediately via invoking Theorem 1 with $\alpha = 1/2$. $\qquad\square$

## C.2 Proof of Corollary 2

We first establish the following lemma on the mean(-square)-uniform stability of $A_{\mathrm{SGD-w}}$ in the case of non-smooth convex loss.

**Lemma 9.** *Suppose that the loss function is $\ell(\cdot;\cdot)$ is convex and $G$-Lipschitz with respect to its first argument. Then $A_{\mathrm{SGD-w}}$ has mean-uniform stability such that*

$$
\sup_{S \doteq S'}\mathbb{E}_{A_{\mathrm{SGD-w}}}\left[\|\|A_{\mathrm{SGD-w}}(S) - A_{\mathrm{SGD-w}}(S')\|\|\right] \le 2G\sqrt{\sum_{t=1}^{T}\eta_t^2} + \frac{4G}{N}\sum_{t=1}^{T}\eta_t,
$$

*and has mean-square-uniform stability such that*

$$
\sup_{S \doteq S'}\mathbb{E}_{A_{\mathrm{SGD-w}}}\left[\|A_{\mathrm{SGD-w}}(S) - A_{\mathrm{SGD-w}}(S')\|^2\right] \le 40G^2\sum_{t=1}^{T}\eta_t^2 + \frac{32G^2}{N^2}\left(\sum_{t=1}^{T}\eta_t\right)^2.
$$

*Proof.* Let us define the sequences $\{w_t\}_{t\in[T]}$ and $\{w'_t\}_{t\in[T]}$ that are respectively generated over $S$ and $S'$ via $A_{\mathrm{SGD-w}}$ via sample path $\xi = \{\xi_t\}_{t\in[T]}$. Suppose that $S \doteq S'$ and consider a hitting time variable $t_0 = \inf\{t : z_{\xi_t} \ne z'_{\xi_t}\}$. Let $\beta_t = \beta_t(S, S', \xi) := \mathbf{1}_{\{z_{\xi_t} \ne z'_{\xi_t}\}}$ be the random indication function associated with event $z_{\xi_t} \ne z'_{\xi_t}$. Conditioned on $t_0$, it has been shown by Bassily et al. (2020, Lemma 3.1) that

$$
\|w_t - w'_t\| \le 2G\sqrt{\sum_{\tau=t_0}^{t}\eta_\tau^2} + 4G\sum_{\tau=t_0+1}^{t}\beta_\tau\eta_\tau \le 2G\sqrt{\sum_{\tau=1}^{t}\eta_\tau^2} + 4G\sum_{\tau=1}^{t}\beta_\tau\eta_\tau. \tag{A.5}
$$

Then we can show the following for all $t \le T$

$$
\mathbb{E}_{\xi_{[t]}}\left[\|w_t - w'_t\|\right] \le 2G\sqrt{\sum_{\tau=1}^{t}\eta_\tau^2} + \frac{4G}{N}\sum_{\tau=1}^{t}\eta_\tau,
$$

where we have used the fact that $\{\beta_t\}$ is an i.i.d. sequence of `Bernoulli`$(1/N)$ random variables. The convexity of Euclidean norm leads to

$$
\mathbb{E}_{\xi_{[T]}}\left[\|\bar{w}_T - \bar{w}'_T\|\right] \le \frac{\sum_{t=1}^{T}\mathbb{E}_{\xi_{[t]}}\left[\|w_t - w'_t\|\right]}{T} \le 2G\sqrt{\sum_{t=1}^{T}\eta_t^2} + \frac{4G}{N}\sum_{t=1}^{T}\eta_t,
$$

which is the first desired bound. Similarly, based on the square of the bound equation A.5 we can show that

$$
\begin{aligned}
\mathbb{E}_{\xi_{[t]}}\left[\|w_t - w'_t\|^2\right] &\le \mathbb{E}_{\xi_{[t]}}\left[8G^2\sum_{\tau=1}^{t}\eta_\tau^2 + 32G^2\left(\sum_{\tau=1}^{t}\beta_\tau\eta_\tau\right)^2\right] \\
&= 8G^2\sum_{\tau=1}^{t}\eta_\tau^2 + 32G^2\mathbb{E}_{\xi_{[t]}}\left[\sum_{\tau=1}^{t}\beta_\tau^2\eta_\tau^2 + \sum_{\tau\ne\tau'}\beta_\tau\beta_{\tau'}\eta_\tau\eta_{\tau'}\right] \\
&= 8G^2\sum_{\tau=1}^{t}\eta_\tau^2 + 32G^2\left(\frac{1}{N}\sum_{\tau=1}^{t}\eta_\tau^2 + \frac{1}{N^2}\sum_{\tau\ne\tau'}\eta_\tau\eta_{\tau'}\right) \\
&\le 40G^2\sum_{\tau=1}^{t}\eta_\tau^2 + \frac{32G^2}{N^2}\left(\sum_{\tau=1}^{t}\eta_\tau\right)^2,
\end{aligned}
$$

where we have used $\mathbb{E}_{\xi_t}[\beta_t] = \mathbb{E}_{\xi_t}[\beta_t^2] = \frac{1}{N}$. It follows directly from the convexity of loss that

$$\mathbb{E}_{\xi_{[T]}}\left[\|\bar{w}_T - \bar{w}_T'\|^2\right] \le 40G^2 \sum_{t=1}^{T} \eta_t^2 + \frac{32G^2}{N^2}\left(\sum_{t=1}^{T}\eta_t\right)^2.$$

The proof is concluded. $\qquad\square$

Equipped with Lemma 9, we are now in the position to prove Corollary 2.

*Proof of Corollary 2.* From Lemma 9 we know that $A_{\text{SGD-w}}$ with non-smooth convex loss has mean(-square)-uniform stability with parameters

$$\gamma_{\text{m},N} = 2G\sqrt{\sum_{t=1}^{T}\eta_t^2} + \frac{4G}{N}\sum_{t=1}^{T}\eta_t, \quad \gamma_{\text{m}^2,N} = 40G^2\sum_{t=1}^{T}\eta_t^2 + \frac{32G^2}{N^2}\left(\sum_{t=1}^{T}\eta_t\right)^2.$$

The desired results then follow immediately via invoking Theorem 1 with $\alpha = 1/2$. $\qquad\square$

## C.3 PROOF OF COROLLARY 3

We first establish the following lemma on the mean(-square)-uniform stability of $A_{\text{SGD-w}}$ in the considered non-convex regime.

**Lemma 10.** *Suppose that the loss function is $\ell(\cdot;\cdot)$ is G-Lipschitz and L-smooth with respect to its first argument. Consider $\eta_t \le 1/L$. Let*

$$u_t := \eta_t^2 + 2\eta_t \sum_{\tau=1}^{t-1} \exp\left(L\sum_{i=\tau+1}^{t-1}\eta_i\right)\eta_\tau$$

*for all $t \ge 1$. Then $A_{\text{SGD-w}}$ has mean-uniform stability such that*

$$\sup_{S \doteq S'} \mathbb{E}_{A_{\text{SGD-w}}}\left[\|A_{\text{SGD-w}}(S) - A_{\text{SGD-w}}(S')\|\right] \le \frac{2G}{N}\sum_{t=1}^{T}\exp\left(L\sum_{\tau=t+1}^{T}\eta_\tau\right)\eta_t,$$

*and has mean-square-uniform stability such that*

$$\sup_{S \doteq S'} \mathbb{E}_{A_{\text{SGD-w}}}\left[\|A_{\text{SGD-w}}(S) - A_{\text{SGD-w}}(S')\|^2\right] \le \frac{4G^2}{N}\sum_{t=1}^{T}\exp\left(3L\sum_{\tau=t+1}^{T}\eta_\tau\right)u_t.$$

*Proof.* Let us define the sequences $\{w_t\}_{t\in[T]}$ and $\{w_t'\}_{t\in[T]}$ that are respectively generated over $S$ and $S'$ via $A_{\text{SGD-w}}$ via sample path $\xi = \{\xi_t\}_{t\in[T]}$. Suppose that $S \doteq S'$. Let us consider $\Delta_t := \mathbb{E}_{\xi_{[t]}}[\|w_t - w_t'\|]$. Then based on the arguments of Hardt et al. (2016, Theorem 3.8) we know that with probability $1 - \frac{1}{N}$ over $\xi_t$, $\|w_t - w_t'\| \le (1 + \eta_t L)\|w_{t-1} - w_{t-1}'\|$, and $\|w_t - w_t'\| \le \|w_{t-1} - w_{t-1}'\| + 2G\eta_t$ with probability $\frac{1}{N}$. Therefore we have

$$\begin{aligned}
\Delta_t &\le \left(1 - \frac{1}{N}\right)(1+\eta_t L)\Delta_{t-1} + \frac{1}{N}\left(\Delta_{t-1} + 2G\eta_t\right) \\
&= \left(\left(1 - \frac{1}{N}\right)(1+\eta_t L) + \frac{1}{N}\right)\Delta_{t-1} + \frac{2G\eta_t}{N} \\
&= \left(1 + \left(1 - \frac{1}{N}\right)\eta_t L\right)\Delta_{t-1} + \frac{2G\eta_t}{N} \\
&\le \exp\left(\left(1 - \frac{1}{N}\right)\eta_t L\right)\Delta_{t-1} + \frac{2G\eta_t}{N} \\
&\le \exp\left(\eta_t L\right)\Delta_{t-1} + \frac{2G\eta_t}{N},
\end{aligned}$$

where we have used $1 + x \leq \exp(x)$. Then we can unwind the above recursion form to obtain that for all $t \geq 1$,

$$\Delta_t \leq \sum_{\tau=1}^{t} \left\{ \prod_{i=\tau+1}^{t} \exp(\eta_i L) \right\} \frac{2G\eta_\tau}{N} = \frac{2G}{N} \sum_{\tau=1}^{t} \exp\left( L \sum_{i=\tau+1}^{t} \eta_i \right) \eta_\tau, \tag{A.6}$$

where we have used $\Delta_0 = 0$. The convexity of Euclidean norm leads to

$$\mathbb{E}_{\xi_{[T]}} \left[ \|\bar{w}_T - \bar{w}_T'\| \right] \leq \frac{\sum_{t=1}^{T} \mathbb{E}_{\xi_{[t]}} \left[ \|w_t - w_t'\| \right]}{T} \leq \frac{2G}{N} \sum_{t=1}^{T} \exp\left( L \sum_{\tau=t+1}^{T} \eta_\tau \right) \eta_t,$$

which immediately implies the first desired bound as it holds for all $S \doteq S'$.

To show the mean-square-uniform stability bound, let us consider $\tilde{\Delta}_t := \mathbb{E}_{\xi_{[t]}} \left[ \|w_t - w_t'\|^2 \right]$. Then we can verify that with probability $1 - \frac{1}{N}$ over $\xi_t$, $\|w_t - w_t'\|^2 \leq (1 + \eta_t L)^2 \|w_{t-1} - w_{t-1}'\|^2$, and with probability $\frac{1}{N}$,

$$\|w_t - w_t'\|^2 \leq (\|w_{t-1} - w_{t-1}'\| + 2G\eta_t)^2 = \|w_{t-1} - w_{t-1}'\|^2 + 4G\eta_t \|w_{t-1} - w_{t-1}'\| + 4G^2 \eta_t^2.$$

Therefore we have

$$\tilde{\Delta}_t \leq \left( 1 - \frac{1}{N} \right) (1 + \eta_t L)^2 \tilde{\Delta}_{t-1} + \frac{1}{N} \left( \tilde{\Delta}_{t-1} + 4G\eta_t \Delta_{t-1} + 4G^2 \eta_t^2 \right)$$

$$\leq \left( \left( 1 - \frac{1}{N} \right) (1 + \eta_t L)^2 + \frac{1}{N} \right) \tilde{\Delta}_{t-1} + \frac{4G^2}{N} \left( \underbrace{\eta_t^2 + 2\eta_t \sum_{\tau=1}^{t-1} \exp\left( L \sum_{i=\tau+1}^{t-1} \eta_i \right) \eta_\tau}_{u_t} \right)$$

$$= \left( 1 + \left( 1 - \frac{1}{N} \right) (2\eta_t L + \eta_t^2 L^2) \right) \tilde{\Delta}_{t-1} + \frac{4G^2 u_t}{N}$$

$$\leq \exp\left( \left( 1 - \frac{1}{N} \right) (2\eta_t L + \eta_t^2 L^2) \right) \tilde{\Delta}_{t-1} + \frac{4G^2 u_t}{N}$$

$$\leq \exp\left( 2\eta_t L + \eta_t^2 L^2 \right) \tilde{\Delta}_{t-1} + \frac{4G^2 u_t}{N},$$

where in the second inequality we have used the bound equation A.6 on $\Delta_t$. Recall that $\tilde{\Delta}_0 = 0$. Then we can unwind the above recursion form to obtain

$$\tilde{\Delta}_t \leq \frac{4G^2}{N} \sum_{\tau=1}^{t} \left\{ \prod_{i=\tau+1}^{t} \exp\left( 2\eta_i L + \eta_i^2 L^2 \right) \right\} u_\tau \leq \frac{4G^2}{N} \sum_{\tau=1}^{t} \exp\left( 3L \sum_{i=\tau+1}^{t} \eta_i \right) u_\tau,$$

where we have used $\eta_t \leq 1/L$. It follows immediately from the convexity that

$$\mathbb{E}_{\xi_{[T]}} \left[ \|\bar{w}_T - \bar{w}_T'\|^2 \right] \leq \frac{\sum_{t=1}^{T} \mathbb{E}_{\xi_{[t]}} \left[ \|w_t - w_t'\|^2 \right]}{T} \leq \frac{4G^2}{N} \sum_{t=1}^{T} \exp\left( 3L \sum_{\tau=t+1}^{T} \eta_\tau \right) u_t,$$

which is the second desired bound. The proof is completed. $\qquad \square$

With Lemma 10 in place, we proceed to prove the main result in Corollary 3.

*Proof of Corollary 3.* From Lemma 10 we know that $A_{\mathrm{SGD-w}}$ with smooth non-convex loss has mean(-square)-uniform stability with parameters

$$\gamma_{\mathrm{m},N} = \frac{2G}{N} \sum_{t=1}^{T} \exp\left( L \sum_{\tau=t+1}^{T} \eta_\tau \right) \eta_t, \quad \gamma_{\mathrm{m}^2,N} = \frac{4G^2}{N} \sum_{t=1}^{T} \exp\left( 3L \sum_{\tau=t+1}^{T} \eta_\tau \right) u_t.$$

The desired results then follow immediately via invoking Theorem 1 with $\alpha = 1/2$. $\qquad \square$

---

**Algorithm 3:** `SGD via Without-Replacement Sampling` ($A_{\text{SGD-wo}}$)

---

**Input** : Data set $S = \{z_i\}_{i \in [N]} \overset{\text{i.i.d.}}{\sim} \mathcal{D}^N$, step-sizes $\{\eta_t\}_{t \geq 1}$, #iterations $T$, initialization $w_0$.
**Output**: $\bar{w}_T = \frac{1}{T} \sum_{t \in [T]} w_t$.
**for** $t = 1, 2, ..., T$ **do**
  Uniformly randomly sample an index $\xi_t \in [N]$ *with* or *without* replacement;
  Compute $w_t = \Pi_{\mathcal{W}} (w_{t-1} - \eta_t \nabla_w \ell(w_{t-1}; z_{\xi_t}))$.
**end**

---

## D  AUGMENTED RESULTS FOR SGD UNDER WITHOUT-REPLACEMENT SAMPLING

In this section, we further consider applying our main results in Theorem 1 to the variant of SGD under without-replacement sampling ($A_{\text{SGD-wo}}$), as is outlined in Algorithm 3. For the sake of simplicity and readability, we only consider single-epoch processing with $T \leq N$, and we focus on the case where the loss is convex but non-smooth. The extensions of our analysis to multi-epoch processing, i.e., $T \leq rN$ for some integer $r \geq 1$, and to convex or non-convex smooth loss functions are more or less straightforward and thus are omit.

We start by establishing the following lemma on the mean(-square)-uniform stability of $A_{\text{SGD-wo}}$ which can be easily proved based on the result from Bassily et al. (2020, Lemma 3.1).

**Lemma 11.** *Suppose that the loss function is $\ell(\cdot; \cdot)$ is convex and $G$-Lipschitz with respect to its first argument. Consider $T \leq N$. Then $A_{\text{SGD-wo}}$ has mean-uniform stability such that*

$$\sup_{S \doteq S'} \mathbb{E}_{A_{\text{SGD-wo}}} \left[ \|A_{\text{SGD-wo}}(S) - A_{\text{SGD-wo}}(S')\| \right] \leq \frac{2G}{N} \sum_{t_0=1}^{T} \sqrt{\sum_{t=t_0}^{T} \eta_t^2},$$

*and has mean-square-uniform stability such that*

$$\sup_{S \doteq S'} \mathbb{E}_{A_{\text{SGD-wo}}} \left[ \|A_{\text{SGD-wo}}(S) - A_{\text{SGD-wo}}(S')\|^2 \right] \leq \frac{4G^2}{N} \sum_{t_0=1}^{T} \sum_{t=t_0}^{T} \eta_t^2.$$

*Proof.* Let $\bar{w}_T(S, \xi)$ and $\bar{w}_T(S', \xi)$ respectively be the output generated over $S = \{z_i\}_{i \in [N]}$ and $S' = \{z'_i\}_{i \in [N]}$ by $A_{\text{SGD-wo}}$ via sample path $\xi = \{\xi_t\}_{t \in [T]}$. Recall that $T \leq N$. Let us define a hitting time variable $t_0 = \inf\{t : z_{\xi_t} \neq z'_{\xi_t}\}$. Since $S \doteq S'$, the uniform randomness of $\xi_t$ implies that

$$\mathbb{P}(t_0 = j) = \frac{1}{N}, \quad j \in [N].$$

Given $t \in [T]$, it follows from Bassily et al. (2020, Lemma 3.1) that

$$\|w_t - w'_t\|^2 \leq 4G^2 \sum_{\tau=t_0}^{t} \eta_\tau^2.$$

Then we have

$$\mathbb{E}_{\xi_{[t]}} \left[ \|w_t - w'_t\|^2 \right] \leq \frac{4G^2}{N} \sum_{t_0=1}^{t} \sum_{\tau=t_0}^{t} \eta_\tau^2 \leq \frac{4G^2}{N} \sum_{t_0=1}^{T} \sum_{\tau=t_0}^{T} \eta_\tau^2.$$

The convexity of squared Euclidean norm leads to

$$\mathbb{E}_{\xi_{[T]}} \left[ \|\bar{w}_T - \bar{w}'_T\|^2 \right] \leq \frac{\sum_{t=1}^{T} \mathbb{E}_{\xi_{[t]}} \left[ \|w_t - w'_t\|^2 \right]}{T} \leq \frac{4G^2}{N} \sum_{t_0=1}^{T} \sum_{t=t_0}^{T} \eta_t^2.$$

Similarly we can show

$$\mathbb{E}_{\xi_{[T]}} \left[ \|\bar{w}_T - \bar{w}'_T\| \right] \leq \frac{2G}{N} \sum_{t_0=1}^{T} \sqrt{\sum_{t=t_0}^{T} \eta_t^2}.$$

The proof is completed. $\qquad \square$

The following result is a direct consequence of Theorem 1 when invoking Algorithm 1 to $A_{\text{SGD-wo}}$ with non-smooth convex loss.

**Corollary 5.** *Suppose that the loss function is $\ell(\cdot; \cdot)$ is convex and G-Lipschitz with respect to its first argument, and is bounded in the range of $[0, M]$. Consider Algorithm 1 specified to $A_{\text{SGD-wo}}$ with $T \leq N$. Then for any $\delta \in (0, 1)$ and $K \geq 2 \log(\frac{4}{\delta})$, with probability at least $1 - \delta$ over the randomness of $S$ and $\{A_{\text{SGD-wo},k}\}_{k \in [K]}$, the output of $A_{\text{SGD-wo}}$ satisfies*

$$|R(A_{\text{SGD-wo},k^*}(S_{k^*})) - R_S(A_{\text{SGD-wo},k^*}(S_{k^*}))|$$

$$\lesssim G^2 \sqrt{\frac{1}{N} \sum_{t_0=1}^{T} \sum_{t=t_0}^{T} \eta_t^2} + \frac{G^2}{N} \sum_{t_0=1}^{T} \sqrt{\sum_{t=t_0}^{T} \eta_t^2} + M \sqrt{\frac{\log(K/\delta)}{N}}.$$

*Proof.* From Lemma 11 we know that $A_{\text{SGD-wo}}$ has mean(-square)-uniform stability with parameters

$$\gamma_{\text{m},N} = \frac{2G}{N} \sum_{t_0=1}^{T} \sqrt{\sum_{t=t_0}^{T} \eta_t^2}, \quad \gamma_{\text{m}^2,N} = \frac{4G^2}{N} \sum_{t_0=1}^{T} \sum_{t=t_0}^{T} \eta_t^2.$$

The results then follow immediately via invoking Theorem 1 with $\alpha = 1/2$. $\qquad\square$

**Remark 9.** *Specially for constant learning rates $\eta_t \equiv \eta$, Corollary 5 admits a high probability generalization bound of scale $\mathcal{O}\left(\frac{\eta T}{\sqrt{N}} + \eta \frac{T\sqrt{T}}{N} + \sqrt{\frac{\log(1/\delta)}{N}}\right)$. For general time varying learning rates, our bound in Corollary 5 still holds with high probability. For example, when $\eta_t \propto \frac{1}{t}$, the generalization bound scales as $\mathcal{O}\left(\sqrt{\frac{\log(T)}{N}} + \frac{\sqrt{T}}{N} + \sqrt{\frac{\log(1/\delta)}{N}}\right)$ with high probability.*

