# OpenReview forum: "Boosting the Confidence of Near-Tight Generalization Bounds for Uniformly Stable Randomized Algorithms"
_ICLR.cc/2022/Conference — ICLR 2022 Submitted_

### Official Review · Reviewer_Ufmg · 2021-10-31

**Correctness:** 3
**Technical Novelty And Significance:** 3
**Empirical Novelty And Significance:** Not applicable
**Recommendation:** 6
**Confidence:** 5

**Main Review:**

Comments:

1. In Corollary 1 2 and 3, Consider Algorithm 1 specified to $A_{\text{SGD-w}}$ is unclear to me, and $\{A_{\text{SGD-w},k}\}$ is not defined. In my understanding, $A_{\text{SGD-w}, k^*} (S_{ k^* } )$ is the output of SGD with subbagging process in Algorithm 1, and Algorithm 2 is seems to redundant here. Did I imiss something ?

2.  In the proof of Corollary 1 2 and 3, the authors should clearly state that they put $\gamma_{m,N/K}$ and $\gamma_{m^2,N/K}$ instead of $\gamma_{m,N}$ and $\gamma_{m^2,N}$ back into Theorem 1, and the bounds are related to $K$ which cannot be ignored. In addition, in the statements of corollaries, setting $\alpha=1/2$â should be added.

3. In Remark 6, choosing $K\asymp\log{(1/\delta)}$ and $\eta_t=1/\sqrt t$, the high probability generalization bound should be dominated by $O\left(\sqrt{\frac{\log{(T)}}{N}}+\frac{\sqrt T+\sqrt{N\log{(1/\delta)}}}{N}\right)$.

4. In Remark 8, the stepsize is chosen to be $\eta_t = O({1\over t^{1+\nu}})$ which seems not right, since the stepsize should satisfy $\sum_{t=1}^\infty \eta_t =\infty$. Is this the artifact of the proof in the nonconvex case?


Mirror comments: Page 4 paragraph 3, $\gamma_{m,N/K}$ and $\gamma_{m^2,N/K}$ have not been defined before. Page 12, in Lemma 4, $h_i$ should be $g_i$.



**Summary Of The Paper:**

 This paper derives near-optimal high probability generalization bounds for randomized learning algorithms. To be specific, the authors establish an improved high probability generalization error bounds over the randomness of data and algorithm via confidence boosting method. The main novelty is they establish the in-expectation first moment generalization bound with mean (square) uniform stability for the randomized algorithm and then use subbagging technique to obtain the high-confidence bounds. In particular, their results can handle time-varying stepsize cases when applied to SGD.  The ideas are motivated by the confidence-boosting/bagging ideas in the literature.

**Summary Of The Review:**

 The paper proposed to use the confidence-boosting/bagging techniques to boost the dependence of learning algorithm from $1/\delta$ to $\log (1/\delta)$.  Specific applications to SGD are derived.   Overall, the paper is well written and the results are new and interesting.

---

> ### Author Response · Authors · 2021-11-15
> **Response to Reviewer Ufmg**
>
> Thank you for your very helpful comments and positive evaluation of our work.
>
> > **Your comment:** In Corollary 1 2 and 3, Consider Algorithm 1 specified to $A_{SGD-w}$ is unclear to me, and $A_{SGD-w, k}$ is not defined. In my understanding, $A_{SGD-w, k^*}(S_{k^*})$ is the output of SGD with subbagging process in Algorithm 1, and Algorithm 2 is seems to redundant here. Did I miss something?
>
> **Our response:** Yes, it is correct that $\\{A_{SGD-w,k}\\}$ denotes the outputs of SGD-w over subsets $\\{S_k\\}$ when applied with the confidence-boosting procedure of Algorithm 1. Per your comment, we have highlighted this definition in the first paragraph of Section 3 in the updated manuscript. In terms of Algorithm 2, it is introduced mainly for clarity purpose because we have simultaneously considered the without-replacement SGD as outlined in Algorithm 3 (see Appendix D).
>
> > **Your comment:** In the proof of Corollary 1 2 and 3, the authors should clearly state that they put $\gamma_{m,N/K}$ and $\gamma_{m^2,N/K}$ instead of $\gamma_{m,N}$ and $\gamma_{m^2,N}$ back into Theorem 1, and the bounds are related to $K$ which cannot be ignored. In addition, in the statements of corollaries, setting $\alpha=1/2$ should be added.
>
> **Our response:** We appreciate your suggestion about clarifying choices of $K$ and $\alpha$ in Corollaries 1, 2 and 3, and have updated the manuscript by explicitly stating $K\asymp \log(1/\delta)$ and $\alpha=1/2$ for these corollaries.
>
> > **Your comment:** In Remark 8, the stepsize is chosen to be $\eta_t=O(\frac{1}{t^{1+\nu}})$ which seems not right, since the stepsize should satisfy $\sum_{t=1}^\infty \eta_t = \infty$. Is this the artifact of the proof in the nonconvex case?
>
> **Our response:** Regarding the choice of time decaying learning rate in Remark 8, there exists a clear trade-off between the generalization (or stability) and optimization performances of SGD. Our original suggested choices of rates in Remark 8 are indeed more attractive for generalization rather than optimization. To ease this concern, we alternatively consider the choice of $\eta_t= \frac{1}{L\nu t}$ with arbitrary $\nu\ge 1$ which has also been considered by Hardt et al. (2016) for analyzing the stability and generalization of non-convex smooth SGD. In this case, it can be verified that the corresponding tail bound in Corollary 3 is of scale $\mathcal{O}(\sqrt{\frac{T^{1/\nu}\log(T)}{\nu N}}+\sqrt{\frac{\log(1/\delta)}{N}})$. Please see Remark 8 of the revised manuscript for the update.
>
> > **Minor comments** on Remark 6 and typos.
>
> **Our response:** Thanks for pointing out typos and other minor issues which have been fixed in the revised manuscript.
>
> ## Reference:
>
> [1] Hardt, et al., Train faster, generalize better: Stability of stochastic gradient descent, ICML, 2016.

---

### Official Review · Reviewer_LQgS · 2021-11-01

**Correctness:** 4
**Technical Novelty And Significance:** 3
**Empirical Novelty And Significance:** 3
**Recommendation:** 6
**Confidence:** 4

**Main Review:**

strengths :
(1.) This paper has nice writing and a clear organization.

(2.) Existing generalization bounds of randomized algorithms are often derived in expectation. Although some work uses the Chernoff bound of Bernoulli random variables to derive the high probability bound, the authors provide improved ones in this paper.

(3.) I checked the proof, which seems to be solid.

Concerns:
(1.) The subbaging process requires an enormous computational complexity. When the authors apply it to SGD, it occurs to me that the high probability generalization bounds are derived for a variant of SGD, not the vanilla SGD.

(2.) Some proof steps in Page 15 between $\mathbb{P}_{S,A_k}(\bar{(\mathcal{E}_1^k)})$ and $\frac{\delta}{2K}$ are confused. The authors can make a detailed explanation to improve their paper’s impact.

(3.) The authors discussed that Lemma 1 is inspired by (Bousquet et al,. 2020). In the proof of Lemma 1, Lemma 4, a basic lemma, is a first moment bound for a summation of random functions, while Theorem 4 in (Bousquet et al,. 2020) are derived for $p$-th moment bound with $p \geq 2$. I do hope the authors can explain why they derive the first moment bound. Is it an essential step to the proof? In my opinion, the $p$-th moment bound can be employed to derive $1/n$-type bound (Klochkov and Zhivotovskiy, 2021). The first moment bound may limit its application to derive sharper bounds.


**Summary Of The Paper:**

This paper establishes near-optimal high probability bound for randomized algorithms with on-average uniform stability. To this end, the authors use a confidence-boosting technique via a subbaging process. The authors then apply the derived generalization bound to SGD and the deterministic uniformly stable algorithm. The results reveal that their bounds improve the results of SGD in (Hardt et al., 2016) and the results in (Bousquet et al,. 2020) up to a $\log (N)$ term.

**Summary Of The Review:**

Overall, the authors proposed an interesting idea to improve the stability-based generalization bound for randomized algorithms. I thus tend to comment for acceptance.

---

> ### Author Response · Authors · 2021-11-15
> **Response to Reviewer LQgS**
>
> We thank the reviewer for the detailed review and positive feedback.
>
> > **Your comment:** The subbaging process requires an enormous computational complexity. When the authors apply it to SGD, it occurs to me that the high probability generalization bounds are derived for a variant of SGD, not the vanilla SGD.
>
>
> **Our response:** Concerning the tail bounds for vanilla SGD under on-average uniform stability, it is a very thrilling question but still remains open. The confidence-boosting technique does not allow us to obtain the desired near-tight bounds for SGD over the entire training set. We leave the full understanding of this challenging problem for future investigation.
>
> > **Your comment:** Some proof steps in Page 15 between $\mathbb{P}_{S, A_k} \left( \bar{\mathcal{E}^k_1}\right)$ and $ \frac{\delta}{2K}$ are confused. The authors can make a detailed explanation to improve their paper’s impact.
>
> **Our response:** Per your suggestion, we have made more detailed explanations on the motivation and proof steps of the inequality $\mathbb{P}_{S, A_k} \left(\bar{\mathcal{E}^k_1}\right) \le \frac{\delta}{2K}$. Please see Page 15 of the revised manuscript for the update around that inequality.
>
> > **Your comment:** The authors discussed that Lemma 1 is inspired by (Bousquet et al,. 2020). In the proof of Lemma 1, Lemma 4, a basic lemma, is a first moment bound for a summation of random functions, while Theorem 4 in (Bousquet et al,. 2020) are derived for $p$-th moment bound with $p\ge 2$. I do hope the authors can explain why they derive the first moment bound. Is it an essential step to the proof? In my opinion, the $p$-th moment bound can be employed to derive $1/n$-type bound (Klochkov and Zhivotovskiy, 2021). The first moment bound may limit its application to derive sharper bounds.
>
> **Our response:** In terms of the motivation of deriving the first moment bound, we would like to highlight that there are two reasons that make the $p$-th moment inequality in Theorem 4 of Bousquet et al. (2020) less suitable for our analysis for randomized algorithms: First and foremost, as we mentioned below Equation (3) that while being sharp in the dependence on sample size, the resulting high probability bound from the $p$-th moment inequality only holds in expectation with respect to the internal randomness of algorithm. Second, since there is a $\log(N)$ factor involved in that $p$-th moment inequality, its implied first-moment inequality turns out to be sub-optimal in view of our Lemma 1 which suggests that such a logarithmic factor can actually be avoided at least for the first and second moment bounds. We have further highlighted the tightness of our first moment bound in Remark 1 of the revised manuscript.

---

### Official Review · Reviewer_jM1w · 2021-11-03

**Correctness:** 4
**Technical Novelty And Significance:** 2
**Empirical Novelty And Significance:** Not applicable
**Recommendation:** 6
**Confidence:** 2

**Main Review:**

The paper makes progress towards answering, for randomized
algorithms with on-average uniform stability, such as stochastic gradient descent
(SGD) with time decaying learning rates, wheter
these deviation bounds still hold with high confidence over the internal randomness of algorithm.

However clarity can be improved in showing how far paper's results are from lower bounds.

**Summary Of The Paper:**

The paper first establishes an in-expectation first moment generalization error bound for
randomized learning algorithm with on-average uniform stability, based on which
it then shows that a properly designed subbagging process leads to near-tight high
probability generalization bounds over the randomness of data and algorithm. It
further substantializes these generic results to SGD to derive improved high probability generalization bounds for convex or non-convex optimization with natural
time decaying learning rates, which have not been possible to prove with the existing uniform stability results.

**Summary Of The Review:**

The paper makes progress towards answering, for randomized
algorithms with on-average uniform stability, such as stochastic gradient descent
(SGD) with time decaying learning rates, wheter
these deviation bounds still hold with high confidence over the internal randomness of algorithm.

However clarity can be improved in showing how far paper's results are from lower bounds.

---

> ### Author Response · Authors · 2021-11-15
> **Response to Reviewer jM1w**
>
> Thank you for the overall positive feedback and helpful comments.
>
> > **Your comment:** However clarity can be improved in showing how far paper's results are from lower bounds.
>
> **Our response:** Regarding the tightness of Theorem 1, we would like to clarify that the confidence term $\sqrt{\frac{\log(1/\delta)}{N}}$ is necessary since even for an algorithm that outputs a fixed function this is the optimal bound that one can expect on the sampling error. The on-average uniform stability terms $\gamma_{m,\frac{N}{K}}$ and $\gamma_{m^2,\frac{N}{K}}$ are also near-tight as the algorithm output can change arbitrarily with respect to these quantities, at least in expectation. Per your suggestion, we have expanded Remark 3 to further clarity the optimality of our results.

---

### Official Review · Reviewer_F7yX · 2021-11-04

**Correctness:** 4
**Technical Novelty And Significance:** 4
**Empirical Novelty And Significance:** Not applicable
**Recommendation:** 8
**Confidence:** 3

**Main Review:**

Strengths:

1. The problem of showing tighter generalization bounds for uniformly stable algorithms has seen exciting work recently, and the paper makes solid contributions to this directions.

2. The paper is quite well-written. It discusses the prior work properly in context, provides sufficient intuition for their results and is generally quite easy to read.

3. It's nice that even thought the framework is developed for randomized algorithms, it yields some tighter bounds for previous deterministic settings as well. This makes it seem a bit less specialized.

Weaknesses:

1. Continuing from the previous point, one could argue that the results in the paper are a little bit specialized. They hold for a specific regime of SGD, and though the authors do provide context about the importance of the results, perhaps a bit more discussion, discussion of the open problem from previous work and how the paper is concretely making substantial progress on it would be helpful.

2. It's a bit unsatisfying that the bounds don't hold for SGD itself, but instead for the confidence boosted procedure which Alg. 1 develops. Do the authors believe that this is necessary? Is it possible to show the high-probability results for SGD itself?

**Summary Of The Paper:**

The paper considers the problem of showing tight generalization bounds for randomized uniformly stable algorithms. The key contributions are the following:

1. The paper uses the framework of confidence-boosting to develop a generic procedure for amplifying the success probability in the generalization bound for a uniformly stable algorithm.

2. The paper then instantiates this bound for the important case of SGD, obtaining generalization bounds that hold in step sizes regime not covered by previous work.

3. Finally, the paper shows that though the framework is developed for randomized algorithms it obtains tighter bounds for deterministic algorithms as well.

**Summary Of The Review:**

Overall, this is a well-written paper which uses interesting techniques and makes decent progress on an interesting problem, and hence I am in favor of acceptance.

---

> ### Author Response · Authors · 2021-11-15
> **Response to Reviewer F7yX**
>
> Thank you for your insightful review and appreciation of our work.
>
> > **Your comment:** Continuing from the previous point, one could argue that the results in the paper are a little bit specialized. They hold for a specific regime of SGD, and though the authors do provide context about the importance of the results, perhaps a bit more discussion, discussion of the open problem from previous work and how the paper is concretely making substantial progress on it would be helpful.
>
> **Our response:**
>
> 1. We appreciate your suggestion about adding a bit more discussion on the open problem that we address in this work. Per your comment, in the updated manuscript, we explicitly state at the end of Section 1.1 the open problem and motivation of study. To be more specific, we point out there that by far it still remains open to know if the existing deviation bounds for uniformly stable algorithms can hold with high confidence for randomized algorithms with on-average uniform stability (such as SGD with decaying learning rates). The main motivation of our study is to address this open issue and derive sharper high-probability generalization bounds for randomized algorithms that hold jointly over the randomness of data and algorithm, based on the relatively weaker notion of on-average uniform stability. As highlighted in the first paragraph of Section 1.2, the fundamental progress we made in this work is revealing that the confidence-boosting trick yields near-tight high probability generalization bounds for uniformly stable randomized learning algorithms.
>
> 2. Regarding the applicability of our theory beyond SGD, in view of the work of Charles \& Papailiopoulos (2018), it is expected that our proposed confidence-boosting meta algorithm can yield near-tight generalization bounds for randomized optimization algorithms such as randomized coordinate descent (RCD) and the stochastic variance reduced gradient method (SVRG), in both the Polyak-Lojasiewicz non-convex and the strongly convex regimes.
>
> > **Your comment:** It's a bit unsatisfying that the bounds don't hold for SGD itself, but instead for the confidence boosted procedure which Alg. 1 develops. Do the authors believe that this is necessary? Is it possible to show the high-probability results for SGD itself?
>
> **Our response:** The natural question of deriving near-optimal tail bounds for SGD itself under on-average uniform stability conditions is thrilling, but so far our confidence-boosting technique does not allow us to resolve this open problem completely. We do believe that developing tighter martingale concentration bounds for uniformly stable randomized algorithms would be a vital step towards fully resolving this challenging problem.
>
> ## Reference:
>
> [1] Charles \& Papailiopoulos, Stability and generalization of learning algorithms that converge to global optima, ICML, 2018.

---

> > ### Comment · Reviewer_F7yX · 2021-11-30
> > **Thank you for your response**
> >
> > I thank the authors for the detailed response. I'm happy with the changes made in the introduction, and will keep my original score of 8.

---

### Author Response · Authors · 2021-11-15
**General Response**

We sincerely thank all the referees for their appreciation of our work and providing highly constructive comments for improvement. We have carefully revised the manuscript based on the initial reviews. The following is a summary of major changes:

 - At the end of Section 1.1, we explicitly state the open problem and motivation of study.

 - In Remark 1, we highlight the advantage of the first moment bound in Lemma 1 over that implied by the $p$-th moment inequality of Bousquet et al. (2020, Theorem 4).

- In Remark 3, we add a short remark on the tightness of our main results.

- In Remark 8, we alternatively consider the choice of step-size $\eta_t= \frac{1}{L\nu t}$ with arbitrary $\nu\ge 1$ to illustrate the bound of Corollary 3 for SGD with time decaying learning rate.

We hope that the given concerns have been addressed satisfactorily in the revised manuscript and the point-by-point responses to the reviewers' comments.

---

### Decision · Program_Chairs · 2022-01-20

**Decision:**

Reject

**Comment:**

Confidence boosting via aggregating multiple run of algorithms has been used before. The main result of the paper relies on a generic confidence boosting trick. The authors for instance cite Shalev-Schwartz et al 2010 theorem 26 in remark 4 of their paper and correctly point out that for deterministic algorithms like ERM one can use that for confidence boosting. While that theorem there is proved for excess risk and for deterministic algorithms, the main idea there to me seems like what is used in the authors paper as well.

The basic idea:
Property A holds in expectation, Hence use Markov inequality to get a low grade probability version of it in each of the K pieces
Now probability that at least one of the pieces is good is high since each piece is independent of the other
Finally aggregate with simple concentration with union bound.

In Shalev-Schwartz et al 2010 this is done with property being excess risk, here it is done with generalization error.

(Oh and I should add, the fact that the algorithm is randomized does not affect this line of reasoning as long as we use fresh randomness for each of the K blocks).

Now the missing piece covered is that on-average stability implies generalization in expectation. But isn’t this already known to be true in the stability literature?

To me it seems like the main technical contribution of the paper is not as novel. Further, as one of the reviewers points out, the main goal should be to prove high probability guarantee for the algorithm popularly used like SGD not the confidence boosted version of it.

None the less, it seems like the application of the result to SGD seems interesting and somewhat new.

I am reluctant to propose an accept here.